# TELECOMTS: A MULTI-MODAL OBSERVABILITY DATASET FOR TIME SERIES AND LANGUAGE ANALYSIS

## ABSTRACT

Modern enterprises generate vast streams of time series metrics when monitoring complex systems, known as observability data. Unlike conventional time series from domains such as weather, observability data are zero-inflated, highly stochastic, and exhibit minimal temporal structure. Despite their importance, observability datasets are underrepresented in public benchmarks due to proprietary restrictions. Existing datasets are often anonymized and normalized, removing scale information and limiting their use for tasks beyond forecasting, such as anomaly detection, root-cause analysis, and multi-modal reasoning. To address this gap, we introduce `TelecomTS`, a large-scale observability dataset derived from a 5G telecommunications network. `TelecomTS` features heterogeneous, de-anonymized covariates with explicit scale information and supports a suite of downstream tasks, including anomaly detection, root-cause analysis, and a question-answering benchmark requiring multi-modal reasoning. Benchmarking state-of-the-art time series, language, and reasoning models reveals that existing approaches struggle with the abrupt, noisy, and high-variance dynamics of observability data. Our experiments also underscore the importance of preserving covariates' absolute scale, emphasizing the need for foundation time series models that natively leverage scale information for practical observability applications[1].

## 1 INTRODUCTION

Time series data is ubiquitous across fields such as weather, finance, and energy systems Hu et al. (2025); Kong et al. (2025a); Farahani et al. (2023); Noshad et al. (2019); Fassois & Sakellariou (2009). One particular domain that has been under-studied but is now garnering increasing attention is the observability domain, which focuses on analyzing time series metrics generated by monitoring complex systems to detect anomalies, diagnose issues, and maintain system health Cohen et al. (2025); Palaskar et al. (2024). This observability data includes CPU and memory utilization, network throughput, request latency, error rates, and disk I/O, each offering critical insight into the state and performance of the system.

Compared to data found in weather or other commonly studied time series domains, observability data is fundamentally different and poses unique modeling challenges due to its distinctive characteristics. First, it is highly zero-inflated: many metrics track infrequent events, such as bursts of user traffic, resulting in sparse time series dominated by zeros punctuated by informative spikes. Second, it displays highly dynamic patterns characterized by frequent, abrupt transitions that are challenging to model, whereas current standard time series datasets are much less spiky. Finally, observability data is highly stochastic, with metrics often appearing irregular and exhibiting minimal discernible temporal structure Datadog (2024); Cohen et al. (2025).

Despite their importance and the challenges they present, these types of time series data remain relatively understudied in the time series literature. This gap can be attributed to several factors: (1) the lack of publicly available datasets due to the proprietary nature of observability data, (2) anonymization in the few existing datasets, which obscures both the identity of the metrics and vital information such as their absolute scale; and (3) the limited utility of forecasting, as the erratic, zero-inflated nature of these series makes forecasting less critical, shifting the focus instead toward

---

[1] https://anonymous.4open.science/r/TelecomTS_Benchmark-72AF

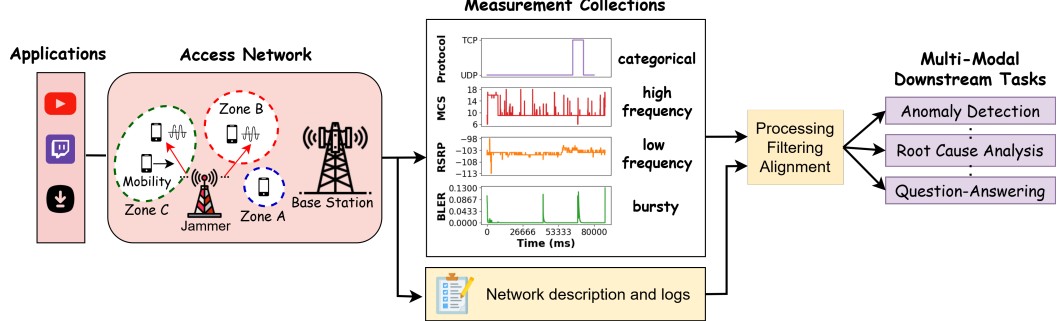

Figure 1: An overview of `TelecomTS`, illustrating its data curation pipeline, covariate characteristics, and the range of supported multi-modal downstream tasks.

anomaly detection, root-cause analysis, and multi-modal reasoning, which are tasks that demand dedicated datasets and systematic curation efforts.

Our paper aims to bridge this gap by introducing `TelecomTS`, a large-scale observability dataset focused on the telecommunications domain. An overview of `TelecomTS` can be found in Fig. 1. Compared to prior datasets, `TelecomTS` differs in two major ways:

**1. Heterogeneous, and de-anonymized covariates with scale information:** Built from extensive data collection on a 5G network, `TelecomTS` contains over 1M observations of KPIs across all layers of the protocol stack Forouzan & Fegan (2002). The dataset captures categorical covariates from dynamically changing communication protocols along with mixed data types metrics (integers and floating-point variables with diverse ranges and distinct statistical distributions), reflecting the inherently heterogeneous nature of observability data. Crucially, it provides full visibility and scale information for each covariates, thus enabling the design of meaningful downstream tasks grounded in operational semantics, as well as facilitating investigation of the impact of scale information and normalization strategies in observability settings.

**2. Comprehensive suite of downstream tasks:** Since observability applications extend beyond simple forecasting, `TelecomTS` incorporates a diverse set of anomalies, including real anomalies generated via controlled jamming signals and synthetically curated rare events grounded in scholarly descriptions of real-world failures. This enables native support for tasks such as anomaly detection and root cause analysis. In addition, we provide a question-answering (Q&A) benchmark that combines temporal reasoning with domain-specific questions tied to the semantics of the network observability environment.

By benchmarking foundation time series models, language models, reasoning models, and lightweight time series baselines on `TelecomTS`, we demonstrate that state-of-the-art approaches consistently struggle with the abrupt, noisy, and high-variance dynamics of this observability data. These challenges manifest as elevated false positives in anomaly detection, misdiagnosed root causes, and poor performance on time series Q&A tasks. Our experiments further underscore the pivotal role of preserving covariates' absolute scale in improving downstream task performance, highlighting the need for developing time series foundation models that explicitly accommodate scale information to achieve superior performance in practical observability applications.

## 2 RELATED WORK

**Time Series Foundation Models**. Recent advances in time series foundation models Ansari et al. (2024); Woo et al. (2024b); Das et al. (2024) have demonstrated strong zero-shot performance across time series benchmarks such as GIFT-EVAL Aksu et al. (2024). Trained on large, multi-domain time series corpora, these models have emerged as a dominant paradigm for time series learning, as they remove the need for extensive task-specific training. This, in turn, facilitates cost-effective zero-shot inference and reduces fine-tuning requirements to a minimum when downstream task adaptation is needed Kottapalli et al. (2025); Faw et al. (2025).

**Time Series Datasets**. The datasets used to train these foundation models span a wide range of domains, including energy Zhou et al. (2021), climate Mouatadid et al. (2024), sales, and transporta-

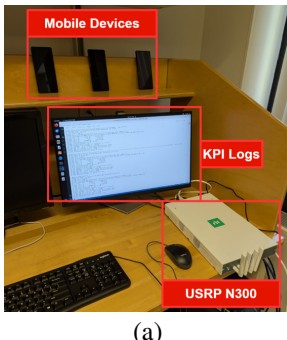 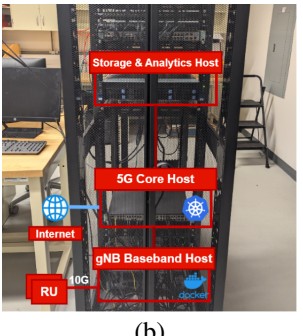 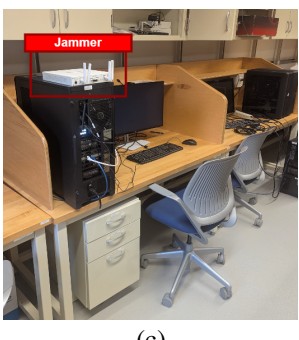

(a)              (b)              (c)

Figure 2: Overview of the 5G wireless network used for data collection: (a) mobile devices used to generate network traffic; (b) server infrastructure hosting the core network, and base-station workloads; (c) programmable jammer used to introduce controlled over-the-air interference.

tion Makridakis et al. (2022); Jiang et al. (2024). In addition, meta-datasets that aggregate multiple sources have been introduced, most notably Monash Godahewa et al. (2021) and the Time Series Pile Goswami et al. (2024). Despite their broad coverage, these datasets largely exclude observability data, which remains scarce in the literature due to its proprietary nature (e.g., customer traffic data from cloud or network operators) Xie et al. (2025); Qureshi et al. (2023).

**Observability Datasets**. Given this gap, time series foundation models have been shown to underperform on observability data Toner et al. (2025); Palaskar et al. (2024). This led to a growing effort in the community to bridge the gap by curating and publishing observability-focused datasets. A notable recent contribution in this direction is the BOOM dataset Datadog (2024); Cohen et al. (2025), which consists of real-world metrics collected from Datadog. BOOM captures a wide spectrum of observability signals from distributed systems, including infrastructure, database, and security.

**Lingering Gaps**. Despite the advancements introduced by the BOOM dataset, several limitations remain. First, the data is anonymized, providing no information about the actual time series variates. Second, the dataset is normalized to preserve privacy. These constraints have multiple consequences: (1) anonymization limits the ability to augment these time series observations with tasks beyond forecasting, such as anomaly detection, multi-modal reasoning, and question-answering, as has been done in other domains such as finance and weather Dau et al. (2018); Liu et al. (2024a); Chen et al. (2025); (2) normalization and loss of absolute scale obscure critical information, particularly in observability contexts where metric magnitudes (e.g., CPU load) are essential for downstream tasks like anomaly detection Lin et al. (2024); and (3) BOOM focuses solely on numerical time-series, providing no support for tasks that combine time-series data with natural language. Consequently, there remains a strong need for de-anonymized observability datasets that provide fully detailed metrics and support a broad range of downstream tasks, including anomaly detection, root-cause analysis, and natural language question-answering.

## 3 TELECOMTS DATASET

In this section, we explore the curation process of `TelecomTS` in detail, covering the raw data, anomalies, and question-answering. We conclude by presenting an overview of the dataset's statistics. A detailed comparison of the covariate behaviors in `TelecomTS` versus those commonly found in the literature is provided in Appendix A.

### 3.1 RAW DATA COLLECTION

**5G Network**. Since telecommunications data is usually proprietary to network operators, comprehensive open-source datasets in this field remain limited. For this reason, we collect our networking data using a 5G wireless network that we developed in our lab, hence free of any privacy concerns. The setup consisted of a single base station (gNB) connected to a full-stack 5G core network serving as the gateway to the Internet. A mobile device was connected to the network and used to generate live traffic using real-world applications such as YouTube, Twitch, and file downloads. The overall architecture of the network is illustrated in Fig. 2(a) and Fig. 2(b).

**Measurement Collection**. During each user connection to the internet, 18 KPIs were recorded from both the base station and the device at 100 ms resolution. Since the data collection was conducted across two separate traces, each capturing different types of KPIs, a time misalignment offset was introduced between the traces. To correct for this offset, we selected two highly correlated KPIs from each trace and applied a histogram-matching technique. Specifically, we aligned the two metrics by temporally shifting one relative to the other and finding the time offset that minimizes the Kullback–Leibler (KL) divergence between their histograms.

**Zoning for Radio Conditions**. To introduce spatial variability in our data, the lab environment was divided into three zones based on distance from the base station. Zone A (0–3 m) provided strong signal quality; Zone B (3–6 m) reflected moderate signal quality, and Zone C (>6 m) delivered weak signal environments.

**Mobility and Congestion**. Data was also collected under static and mobile device conditions to reflect realistic user mobility. Congestion scenarios were emulated by introducing secondary devices that generated heavy traffic, thus creating resource contention representative of high-load environments.

Additional information on the network setup, collected KPIs, and layout is available in Appendix B.

## 3.2 ANOMALIES CURATION

Popular anomaly detection datasets (e.g., UCR Dau et al. (2018)) often contain a combination of real and synthetic anomalies. This is because real anomalies are inherently rare and difficult to capture at scale. Following a similar methodology, `TelecomTS` integrates both real anomalies and a principled approach for generating synthetic ones, as shown in Fig. 3.

**Real anomalies.** In our setup, an adversarial jammer is employed to emit electromagnetic signals on the same frequencies used by mobile devices, thereby interfering with their transmissions. The jammer alternates between active and idle periods; during its active phase, it disrupts network commu-

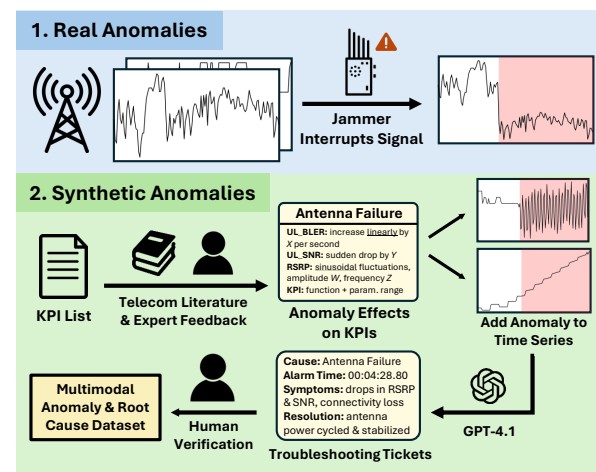

Figure 3: An overview of the anomalies curation process.

nication, causing packet loss and anomalous behavior throughout the network. A visual illustration of this setup is shown in Fig. 2(c), and additional details on the jamming configurations are provided in Appendix B.

**Synthethic Anomalies**. Going beyond real anomalies, we extend our dataset with a comprehensive set of synthetic anomalies. A key challenge in generating synthetic anomalies is ensuring that they faithfully mimic the characteristics of rare, real-world network anomalies rather than simply introducing random drops or unsubstantiated perturbations in the time series. To address this, we adopt a principled methodology for realistic synthetic anomaly creation, outlined as follows.

First, we curate a list of ten anomaly types that are known to occur in networked systems, drawing from technical manuals and scholarly material Liu et al. (2023a); Yen et al. (2022); Hasan et al. (2024); Haseeb (2021). For each anomaly type, we identify the corresponding symptoms in terms of KPIs captured in our time series (e.g., sharp drops, gradual linear increases, or abnormal oscillations). These mappings between anomalies and their KPI-level manifestations are then validated through expert review to ensure their relevance.

Once anomalies and their symptoms are defined, we model their occurrence and duration. Following the findings of Maatouk et al. (2024) from large-scale operational networks, we model anomaly durations and inter-arrival times as exponentially distributed random variables with empirically motivated rate parameters. Using these models, we generate synthetic anomalies by manipulating a set of raw observations from the dataset.

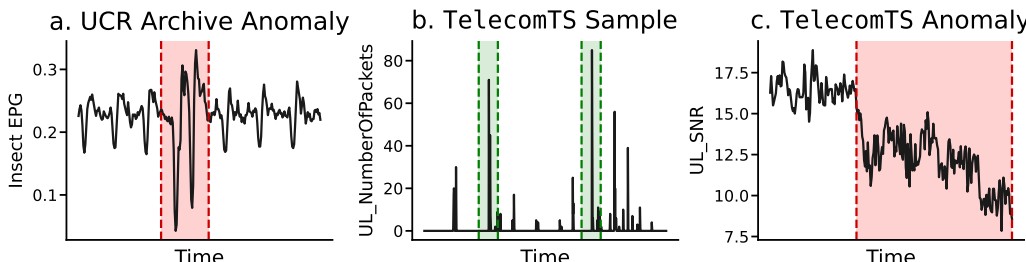

Figure 4: An illustrative difference between UCR Archive Anomaly dataset and the anomalies found in `TelecomTS`. The anomalies found in the former typically manifest as a clear deviation from an otherwise smooth and predictable trend.

Finally, for each synthetic anomalous sample, we generate a textual troubleshooting ticket to enrich the dataset. Each ticket specifies the anomaly type, start and end times, and provides a narrative describing the observed KPIs' behavior during the event. These tickets are produced using GPT-4.1, conditioned on the selected anomaly type, associated symptoms, and temporal boundaries, and are subsequently validated through a human-in-the-loop verification process. This approach mirrors the documentation practices of network operators, where incident reports typically accompany real-world anomalies Maatouk et al. (2025). The complete set of prompts used for this generation, along with all other prompts employed in this work, is included in Appendix D.

**A comparative example**. A key question that arises is how the anomalies in our dataset differ from those found in standard anomaly detection benchmarks such as the UCR archive. To address this, we provide a comparative visualization in Figure 4. Figure 4a displays a typical sample from the UCR anomaly dataset, where anomalies usually appear as distinct deviations from otherwise smooth trends. In contrast, Figure 4b presents a burst in user traffic found in our dataset, which is an abrupt yet entirely normal behavior of observability data. Figure 4c depicts an actual anomaly in our setting, where a sustained shift in the overall trend indicates a true fault rather than fluctuations typical of engineering operations. This comparison highlights a fundamental distinction in our dataset: abrupt changes are often inherent to the dataset and do not necessarily signal anomalous behavior.

## 3.3 QUESTION ANSWERING CURATION

Finally, for an additional multi-modal downstream task, we curate a set of Q&A pairs designed to probe the model's understanding of the time series data. Two families of Q&A are created: the first focuses on qualitative and quantitative aspects to assess a model's ability to reason about inherent statistical and structural properties of the time series. Particularly, for each time series sample and channel, we compute basic metrics such as

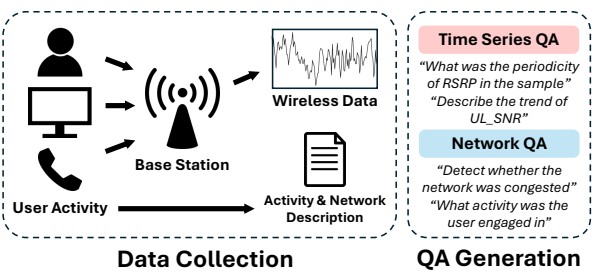

Figure 5: An overview of the Q&A dataset.

mean and variance directly. Periodicity is estimated using a Fourier transform, where the dominant frequency component indicates the primary period. The trend is determined by fitting a linear regression line to the series and evaluating its slope. If the slope exceeds the mean slope plus one standard deviation across all samples, it is labeled as a positive trend; if it falls below the mean minus one standard deviation, it is labeled as a negative trend; otherwise, it is considered to exhibit no prominent trend. The second set of Q&As contains the contextual ground truths of user behavior and network conditions. Particularly, from each sample's metadata, we extract four labels—user activity, mobility state, zone, and congestion status—and prompt GPT-4.1 to generate multiple question–answer templates per label. Then, we randomly sample among these templates to form the final diverse Q&A pairs. An overview of this Q&A is shown in Fig. 5.

### 3.4 OVERALL STATISTICS

All in all, the statistics of our dataset are summarized in Table 1. As shown, the dataset contains over 1 million observations with a wide variety of scenarios, including anomalous behavior, jamming events, and more, all within a 100 ms time-resolution that allows us to capture minute and abrupt system dynamics, aligning with the characteristics of observability data. Additionally, the dataset spans 18 channels comprising diverse data types. For instance, DL_BLER ranges from 0 to 1, TX_Bytes (among five other variables) takes on positive integer values, and RSRP varies between -140 and -60. These characteristics highlight the heterogeneous nature of the dataset. Further details about the dataset can be found in Appendix B. An example of the dataset is provided in Appendix E.

Table 1: Summary statistics of `TelecomTS`.

| Statistic | Category | Count |
|---|---|---|
| Total Samples | Normal Observations | 1,020,000 |
| | Anomalous Observations | 120,000 |
| Time-Resolution | Sampling Rate | 10 Hz |
| Channels | Number of Channels | 18 |
| | Channel Types | 10 float, 6 integers, 2 categorical |
| Synthetic Anomalies | Distinct Anomaly Types | 10 |
| Jamming | Jamming Present Observations | 30,000 |
| Traffic Types | YouTube | 380,000 |
| | Twitch | 380,000 |
| | File Download | 380,000 |
| Mobility | No Mobility | 1,110,000 |
| | In Motion | 30,000 |
| Congestion | Congested Network Observations | 90,000 |
| QnA Types | Time Series QA | 64 |
| | Network QA | 5 |

## 4 EXPERIMENTS

In this section, we conduct comprehensive experiments to evaluate the performance of current state-of-the-art models on the various downstream tasks defined in `TelecomTS`. Our benchmarking includes LLMs, reasoning models, and time series models. Through these experiments, we reveal the performance gap that emerges when existing models are exposed to the complex nature of observability data embodied by our dataset. For all these experiments, the training details are provided in Appendix F.

### 4.1 ANOMALY DETECTION

For this task, we evaluate each model on a randomly selected subset of 1,000 samples from our dataset, where each sample consists of 128 observations containing the full set of KPIs. Results were adapted to ensure a 95%–5% ratio of normal to anomalous instances (including jamming), aligning with class distributions found in existing anomaly detection benchmarks such as UCR Wu & Keogh (2023), and consistent with real-world distributions observed in telecommunications networks Maatouk et al. (2024). Next, for language and reasoning models, we distinguish between two evaluation cases: 1) Without context, where the model was presented with the time series and asked to determine if it is anomalous, and 2) With context, where the model was informed that the data is naturally erratic and can have ups and downs as

Table 2: Anomaly detection precision, recall, and F1 score.

| Model | Precision | Recall | F1 Score |
|---|---|---|---|
| *Large language models* | | | |
| GPT-4.1 (without context) | 0.054 | 1.000 | 0.103 |
| GPT-4.1 (with context) | 0.049 | 0.609 | 0.091 |
| Claude 3.7 Sonnet (without context) | 0.050 | 0.840 | 0.094 |
| Claude 3.7 Sonnet (with context) | 0.054 | 0.860 | 0.101 |
| *Reasoning models* | | | |
| o4-mini (without context) | 0.052 | 1.000 | 0.098 |
| o4-mini (with context) | 0.085 | 0.580 | 0.148 |
| DeepSeek-R1 (without context) | 0.070 | 0.600 | 0.125 |
| DeepSeek-R1 (with context) | 0.072 | 0.470 | 0.125 |
| *Foundation Models* | | | |
| Moment | 0.072 | 0.888 | 0.133 |
| Moirai2 Woo et al. (2024a) | 0.177 | 0.490 | 0.260 |
| Toto Cohen et al. (2025) | 0.3304 | 0.750 | 0.4587 |
| *Time series models* | | | |
| Mantis Feofanov et al. (2025) | 0.640 | 0.800 | **0.711** |
| TimesNet Wu et al. (2023) | 0.118 | 0.535 | 0.194 |
| Autoformer Wu et al. (2022) | 0.066 | 0.690 | 0.121 |
| Non-stationary Transformer Liu et al. (2023b) | 0.347 | 0.655 | 0.453 |
| FEDformer Zhou et al. (2022) | 0.083 | 0.560 | 0.145 |
| Informer Zhou et al. (2021) | 0.123 | 0.690 | 0.208 |

an intrinsic behavior. As for the time series models, we consider both foundation models and popular time series architectures. For foundation models, the backbone is left frozen, and a classification head was used at the end to train for the anomaly detection task.

**Results Analysis.** As shown in Table 2, models like GPT-4.1 and o4-mini exhibit a strong bias toward false positives (i.e., predicting normal samples as anomalous) when no contextual information is provided. Other language models display similar tendencies, though the bias is a bit less severe. This behavior stems from the inherent characteristics of our dataset: abrupt fluctuations are common

in observational data, leading models to misinterpret erratic but normal behavior as anomalous. To illustrate this challenge, we provide a failure case shared across all evaluated models in Fig. 6. As shown, an increase in TX_Bytes, a typical pattern observed during streaming applications, triggers a false positive anomaly prediction, irrespective of the behavior of other channels. This highlights the difficulty these models face in handling naturally abrupt, yet normal, behaviors that are prevalent in practical engineering scenarios.

Next, when additional context is provided, the tendency of models to misclassify naturally fluctuating samples as anomalous is reduced. However, the precision remains low, indicating that models still struggle to distinguish between normal erratic behavior and true anomalies. This challenge extends even to time series foundation models. Despite being pretrained on vast time series datasets, the performance of these models reveals that their learned representations are insufficient to capture this nuanced distinction. Although they surpass the performance of LLMs, the overall performance remains suboptimal, showcasing how current foundation models struggle to handle the complexities of real-world observability data.

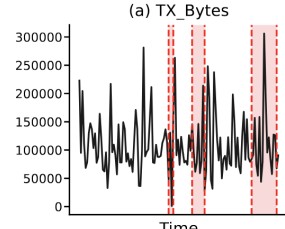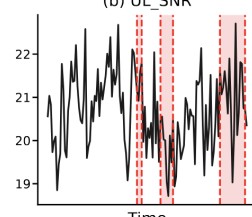

Figure 6: Illustration of a failure case that affected all benchmarked models on this specific sample.

Finally, with respect to the time series models, our results show that most architectures struggle to achieve strong performance on the dataset. A notable exception is Mantis Feofanov et al. (2025), which embeds scale information (specifically, the mean and standard deviation of each patch) into its representations. This design allows the model to remain aware of the absolute values of the time series rather than relying solely on normalized trends, as is the case for other architectures. These findings underscore the importance of preserving scale information and reinforce the value of our dataset, which retains this information, unlike other existing observability datasets in the literature.

## 4.2 ANOMALY DURATION ANALYSIS

In this second task, the objective is to go beyond simple anomaly detection and evaluate the models' ability to localize the duration of anomalies within a sample. To this end, we select only the anomalous samples from the dataset and present them to the models. Given such a sample, language models are prompted to identify the specific segment of the time series where the anomaly occurs. For the time series models, anomaly predictions are generated for each KPI (i.e., covariate) across each timestamp. These predictions are then aggregated via majority voting across variates for each observation to determine the anomalous segment. In both cases, the model outputs are then compared against the ground truth to compute precision, recall, and F1 score. The results are provided in Table 3.

Table 3: Anomaly duration analysis precision, recall, and F1 score.

| Model | Precision | Recall | F1 Score |
|---|---|---|---|
| *Large language models* | | | |
| GPT-4.1 | 0.715 | 0.334 | 0.456 |
| Claude 3.7 Sonnet | 0.697 | 0.292 | 0.412 |
| *Reasoning models* | | | |
| o4-mini | 0.683 | 0.241 | 0.356 |
| DeepSeek-R1 | 0.641 | 0.349 | 0.448 |
| *Foundation Models* | | | |
| Moment | 0.556 | 0.940 | 0.699 |
| Moirai2 | 0.681 | 0.923 | 0.784 |
| Toto | 0.910 | 0.931 | **0.921** |
| *Time series models* | | | |
| Mantis | 0.8734 | 0.9144 | 0.8934 |
| TimesNet | 0.7447 | 0.8661 | 0.8008 |
| Autoformer | 0.6606 | 0.8467 | 0.7422 |
| Non-stationary Transformer | 0.6667 | 0.8279 | 0.7386 |
| FEDformer | 0.6613 | 0.8529 | 0.7450 |
| Informer | 0.6598 | 0.8535 | 0.7442 |

**Results Analysis**. We observe that the models perform relatively well on this task, which suggests that if the model is provided with an already identified anomalous sample, it is more accurate in localizing the anomalous segment within it. All in all, this behavior highlights the importance of dealing with the false positives issue of the detection task, as model performance can be greatly improved once that hurdle of normal fluctuations vs anomaly is dealt with. This further highlights the current limitation of foundation models in practical observability settings and the potential room for improvement.

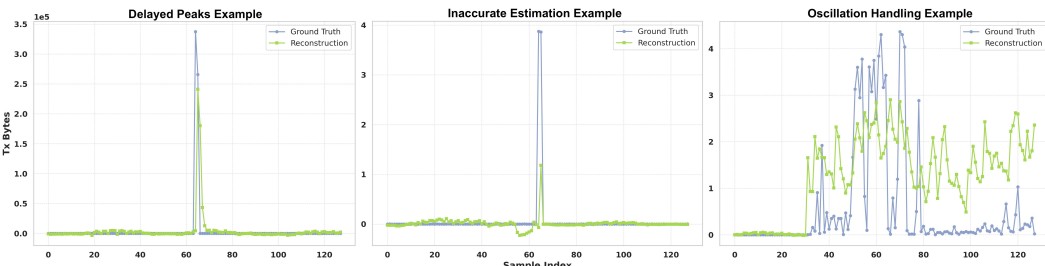

Figure 7: Forecasting results of the highest-performing model (Informer) highlight key challenges: (1) delayed peak predictions, (2) inaccurate magnitude estimation, and (3) difficulty in handling oscillatory patterns.

## 4.3 ROOT CAUSE ANALYSIS

In this task, the objective is to identify the root cause of an anomaly, assessing each model's ability to distinguish between different types of anomalous behaviors. For language models, we present each model with an anomalous sample and ask it to classify the anomaly type among the provided anomaly classes. In the version with context, we also provide the model with information about which KPIs are typically affected by each anomaly type. The results of this evaluation are summarized in Table 4.

**Results Analysis**. Language models perform poorly on this task, struggling to accurately distinguish between different anomaly types, even when provided with contextual information (although performance does improve with context). Time series models equipped with trained classification heads perform better. Notably, Toto performs especially well due to being pretrained on diverse observability data trends, which facilitates effective transfer learning for this task.

Table 4: Accuracy in root cause analysis.

| Model | Accuracy |
|---|---|
| *Large language models* | |
| GPT-4.1 (without context) | 0.215 |
| GPT-4.1 (with context) | 0.227 |
| Claude 3.7-Sonnet (without context) | 0.115 |
| Claude 3.7-Sonnet (with context) | 0.245 |
| *Reasoning models* | |
| o4-mini (without context) | 0.245 |
| o4-mini (with context) | 0.275 |
| DeepSeek-R1 (without context) | 0.145 |
| DeepSeek-R1 (with context) | 0.261 |
| *Foundation Models* | |
| Moment | 0.550 |
| Moirai2 | 0.225 |
| Toto | **0.848** |
| *Time series models* | |
| Mantis | 0.590 |
| TimesNet | 0.685 |
| Autoformer | 0.300 |
| Non-stationary Transformer | 0.520 |
| FEDformer | 0.355 |
| Informer | 0.600 |

## 4.4 FORECASTING

Given that language models are known to perform poorly on forecasting tasks Tan et al. (2024), we focus our analysis on time series models for this evaluation. The performance results of these models are presented in Table 5.

**Results Analysis.** As observed, model performance varies considerably, with some models outperforming others. However, a critical issue that these performance metrics fail to capture is the inherent difficulty of the forecasting task given the sporadic nature of the covariates. In fact, observability data, including our dataset, often exhibits prolonged periods of stable behavior punctuated by sudden spikes, as illustrated in Fig. 7. While models can successfully forecast constant values, they consistently fail to capture peaks—either through delayed detection, inaccurate peak estimation, or difficulty handling oscilla-

Table 5: Accuracy of the models in forecasting.

| Model | MAE | RMSE |
|---|---|---|
| *Foundation models* | | |
| Moment | 0.5435 | 0.7216 |
| Moirai2 | 0.5160 | 0.6988 |
| Toto | 0.4896 | 0.6759 |
| *Time series models* | | |
| Mantis | 0.4578 | 0.6037 |
| TimesNet | 0.1595 | 0.3964 |
| Autoformer | 0.4584 | 0.8948 |
| Non-stationary Transformer | 0.2563 | 0.5608 |
| FEDformer | 0.1702 | 0.4080 |
| Informer | **0.1437** | **0.3586** |

tions. Although the stable periods inflate performance metrics such as MAE (shown in Table 5), this masks the fundamental challenge of forecasting in this environment. Consequently, operating in such environments requires specialized modeling approaches and presents significant challenges that demand novel solutions.

Table 6: Performance of the models on the question-answering task.

| Model | Time series QA | | | | | Network QA | | | |
| | Statistics | | Periodicity | | Trends | Traffic | Mobility | Location | Congestion |
| | $MAE_{min}$ | $MAE_{max}$ | $MAE_{min}$ | $MAE_{max}$ | Acc | Acc | Acc | Acc | Acc |
|---|---|---|---|---|---|---|---|---|---|
| GPT-4.1 | 0.163 | 1588.1 | 57.61 | 93.01 | 16.25 | 44.8 | 53.3 | 29.4 | 49.4 |
| Claude 3.7-Sonnet | 0.093 | 1315.8 | 32.04 | 64.04 | 10.92 | 41.4 | 95.0 | 42.8 | 46.1 |
| o4-mini | 0.027 | 247.1 | 37.21 | 63.15 | 13.37 | 43.3 | 76.7 | 36.7 | 49.4 |
| DeepSeek-R1 | 0.020 | 1542.6 | 50.33 | 61.73 | 13.39 | 35.7 | 98.3 | 33.9 | 48.3 |

## 4.5 QUESTION ANSWERING

As a final task, we evaluate models on question answering by providing natural language questions with time series data to support their responses. Using the Q&A samples outlined in Section 3.3, we design an evaluation pipeline that measures performance using either mean absolute error or accuracy, depending on the type of question. Since the time series component of the Q&A task is structured by covariates, and for ease of presentation, we report results using the KPI that achieves the best and worst performance (in terms of MAE) within the relevant task category. For the network-related Q&A tasks, each model is provided with contextual information regarding the locations zones and overall network configuration. Given the textual component of this task and the lack of strong foundational time series and text models, we restrict our evaluation to language models, as they are currently the only available models capable of processing both natural language and time series data in a unified manner. The results of this evaluation are summarized in Table 6.

**Results Analysis**. The results from this task reveal two key insights. First, in the context of time series Q&A, the model performs well on KPIs that exhibit smooth and stable behavior, where abrupt changes are minimal. However, for more erratic KPIs, particularly TX_Bytes, which naturally exhibits abrupt behavior, the model struggles to make meaningful predictions. This highlights a significant gap in current foundation models ability to analyze statistical characteristics of complex observability signals. Second, with regard to the network-related Q&As, while some models show reasonable performance-especially reasoning ones, they still fall short in effectively linking engineering concepts and the provided contextual knowledge to the underlying time series data. This highlights a critical gap in current models' ability to perform multi-modal reasoning, underscoring the need for models that can more effectively integrate temporal data with textual context.

**Overall Discussions**. All in all, our proposed `TelecomTS` dataset highlights a critical gap between the benchmark performance of current state-of-the-art models and their applicability to real-world observability scenarios. Models that achieve strong results on existing datasets often struggle to generalize to these settings, largely due to the abrupt, noisy, and irregular nature of observability data. In addition, categorical variables, commonly present in observability systems, remain underexplored in the design of time series model architectures, which predominantly focus on numerical variates. Furthermore, the effective encoding of covariates scale information in foundation time series models is still insufficiently studied, despite its demonstrated importance in our experiments on `TelecomTS`.

## 5 CONCLUSIONS

This paper introduced `TelecomTS`, a large-scale, high-resolution, multi-modal dataset designed to bridge the gap between existing time series datasets and the complexities of observability systems. `TelecomTS` comprises over 1 million observations collected from a 5G communication network, incorporating categorical and heterogeneous covariates while capturing the erratic and bursty dynamics characteristic of observability environments. Evaluations of state-of-the-art models—including time series, language, and reasoning models—reveal consistent underperformance on `TelecomTS`. This underperformance stems primarily from their inability to handle the highly erratic patterns characteristic of observability data, as well as their lack of mechanisms to encode and leverage scale information—an aspect that is crucial in such scenarios. These findings underscore the pressing need for more robust and scale-aware time series foundation models capable of effectively operating in complex, real-world observability environments.

**Reproducibility Statement**. To ensure reproducibility, we provide the training and evaluation scripts in the anonymous GitHub found on Page 1. Upon acceptance, we will publicly release the benchmark to support transparent and fair evaluation within the research community. For all LLMs evaluated in our study, we explicitly specify the model and configurations (along with prompts provided in the appendix), allowing experiments to be replicated under the same conditions. Together, these efforts are intended to facilitate rigorous verification of our results.

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

## 5.1 The Use of Large Language Models (LLMs)

Large Language Models (LLMs) he use of LLMs was strictly confined to post-writing editorial tasks. Particularly, they were limited to correcting grammar, punctuation, and spelling errors.

## A Analysis of TelecomTS and Comparison With Existing Datasets

While there exists a plethora of time series datasets, most multivariate datasets lack one or more of the following characteristics crucial for observability data: heterogeneous variates, fine-grained resolution, and categorical variates. Here, we present some commonly used datasets and compare them to TelecomTS.

**Heterogeneous Variates**. We display two case studies that highlight the homogeneity of existing multivariate datasets. The ETTh$_1$ dataset contains data from electrical transformers aggregated on one-hour intervals Zhou et al. (2021). We provide 6 of its 7 variates in Figure 8. As can be seen, the variates share common behavior. Particularly, all variates exhibit monotonic trends and have similar high-frequency dynamics, with spikiness and sharp turning points. For example, by observing the HUFL and LUFL variates, we see a similarity in the peaks and troughs of the variates, and this loosely holds across most variates as well. Moreover, semantically, these variates are also similar since six of them measure six different types of power load. For instance, HUFL measures High Useful Load, LULL measures Low Useless Load, and MUFL measures Middle Useful Load. Only OT, which measures Oil Temperature, is meaningfully different.

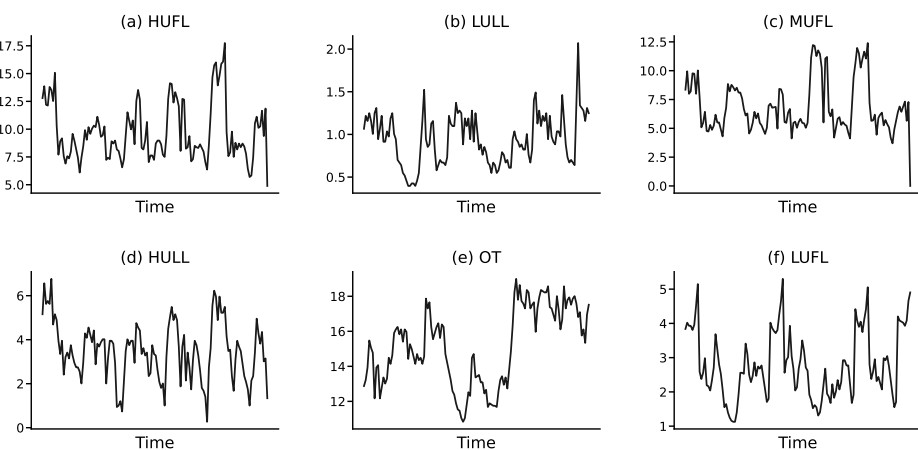

Figure 8: Randomly sampled variates from the ETTh$_1$ dataset.

Next, we observe the MotorImagery dataset that collects EEG data of imagined body movements using an $8 \times 8$ platinum electrode grid. Each of the $64$ sensors corresponds to a variate, and data is recorded every millisecond. While the variates shown in Figure 9 display different trends, we see that they largely behave similarly. The variates lack volatility and exhibit minimal noise and no significant fluctuations or erratic behavior. Particularly, we can see low-frequency structure that spans significant portions of the interval, and we generally have smooth dynamics. Semantically, there is no diversity as all variates represent the same sensor measurement, just at different locations.

The above two examples were some of the many datasets that exhibited homogeneity across their variates. In fact, in our experiments, we selected multivariate datasets containing more than six variates from commonly used time series foundation model datasets, including the Unified Time Series Dataset, LOTSA, and others Liu et al. (2024b); Woo et al. (2024b). We then randomly sampled six variates and time series segments of 128 timestamps. We found that the vast majority of these samples resembled the aforementioned examples, lacking sufficient diversity among their variates.

**Fine-Grained Resolution**. A large portion of existing time series datasets are temporally aggregated or averaged across multiple entities relevant to the scenario at hand. For example, climate metrics

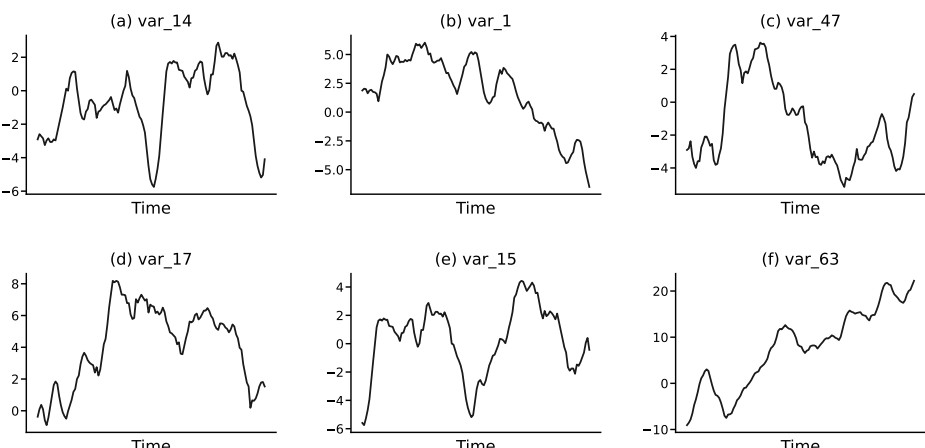

Figure 9: Randomly sampled variates from the MotorImagery dataset.

are typically aggregated at the monthly level, and energy usage data is frequently averaged across different cities. Such aggregation results in smoother and more predictable patterns, rendering these datasets unsuitable for observability applications that demand fine-grained resolution and the capacity to capture erratic, high-variance dynamics.

When it comes to temporal aggregation, the bulk of the datasets used by time series foundation models are recorded hourly, daily, weekly, and monthly. For example, at best, Chronos and TimesFM are trained on time series of 5-minute and 10-minute granularities, respectively Ansari et al. (2024); Liu et al. (2024b). Moirai uses datasets on the second/multi-second granularity, but these comprise only $0.054\%$ of observations Woo et al. (2024b).

To highlight the impact of such resolution on the behavior of the time series, we report the FRED-MD dataset, a macroeconomic dataset comprising 107 variates spanning categories such as consumption, labor, income, interest rates, and other economic indicators McCracken & and (2016). Notably, the data is collected on a monthly frequency, which helps illuminate long-term macroeconomic trends. As seen in Figure 10, this leads to smooth trends, where several variables exhibit strong, positive, and smooth trends, while others display low-frequency fluctuations with minimal abrupt changes. This is far from the erratic and high-variance environments encountered in observability applications. In another example, we report the Weather dataset that contains hourly data on temperature, humidity, wind, and other climate metrics Zhou et al. (2021). Under the hourly frequency, we can see that daily or weekly trends dominate in Figure 11. In particular, DewPointFarenheit and DryBulbCelsius exhibit strong daily fluctuations, which can be too predictable and less relevant to erratic observability dynamics. The other variates also exhibit relatively smooth trends and low variance between consecutive timestamps.

Regarding spatial aggregation, many time series datasets collect data on the city, state, or even country level, which can smooth out less predictable, high-frequency behavior. For example, the COVID Deaths dataset documents daily deaths from the COVID-19 pandemic where each time series corresponds to a whole country Dong et al. (2020). Similarly, the CDC Fluview ILINet captures illness data on the state, regional, and national level Centers for Disease Control and Prevention (2017). Although spatial aggregation and temporal aggregation are often unsuitable for many observability applications, we note that there exists increasing interest in time series datasets with fine-grained spatial resolution due to the increased popularity of spatiotemporal data and distributed sensor deployment Jiang et al. (2024).

**Multi-modal Downstream Tasks**. The lack of multi-modal time series datasets remains a significant bottleneck in the development of capable multimodal time series foundation models. There exists few natively multimodal datasets, most notably Time-MMD, while most datasets retroactively annotate existing time series data, such as in TIME-MQA which annotates datasets from UTSD Liu et al. (2025); Kong et al. (2025b). As a result, many studies in this domain are forced to bootstrap their

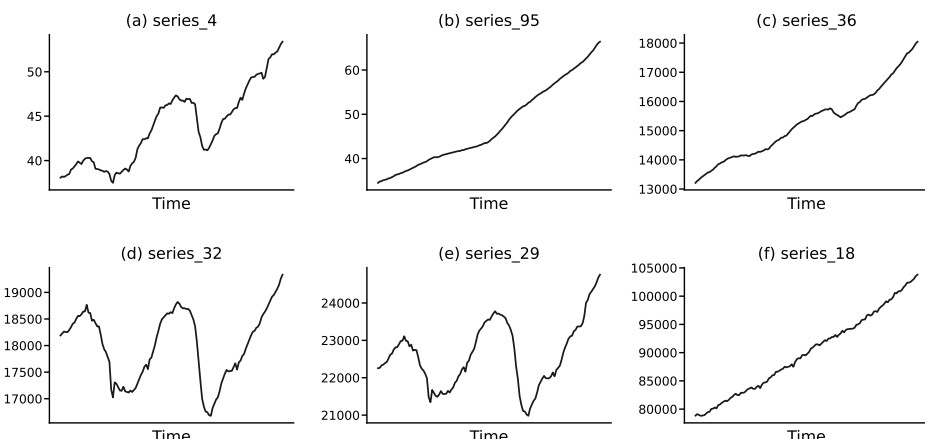

Figure 10: Randomly sampled variates from the FRED-MD dataset.

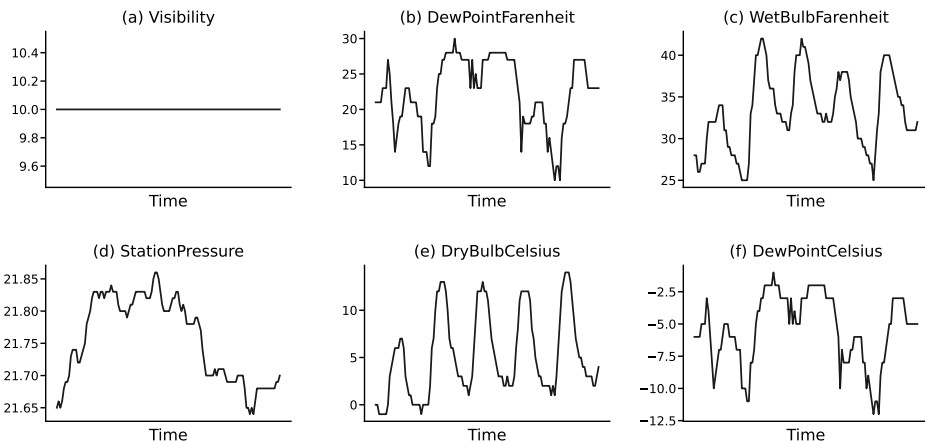

Figure 11: Randomly sampled variates from the WTH dataset.

own datasets. Moreover, the majority of existing datasets are designed solely for forecasting tasks, with limited support for other practical applications such as anomaly detection or root cause analysis.

**Our Dataset**. From Figures 12 and 13, we see that our dataset starkly differs from the previously displayed examples. Firstly, UL_Protocol and DL_Protocol are both categorical variates that exhibit unique temporal and statistical dynamics. Following this, we see that we have high variate diversity. In our numerical data, we have low-frequency variates such as UL_MCS, DL_MCS, and RSRP, which may be less erratic. On the other hand, we have high variance and noisy variates with UL_SNR, UL_NPRB, etc. Even within our high variance variates, we have lots of diversity. We can see that UL_SNR has many sharp turns, frequent spikes and troughs. On the other hand, UL_NPRB is very spiky in one direction and often resets to a baseline value. Moreover, both RX_Bytes and TX_Bytes exhibit sporadic spikes at lower frequencies, typically corresponding to specific events—such as bursts in downloaded data. These observations exemplify the importance of fine-grained data, as such unpredictable spikes are not averaged or aggregated out at our 100ms time-scale. Finally, beyond the behavior of the covariates, it is important to note that our variates span multiple data types—ranging from integers (e.g., TX_Bytes) to floating-point values such UL_BLER—covering distinct range of values.

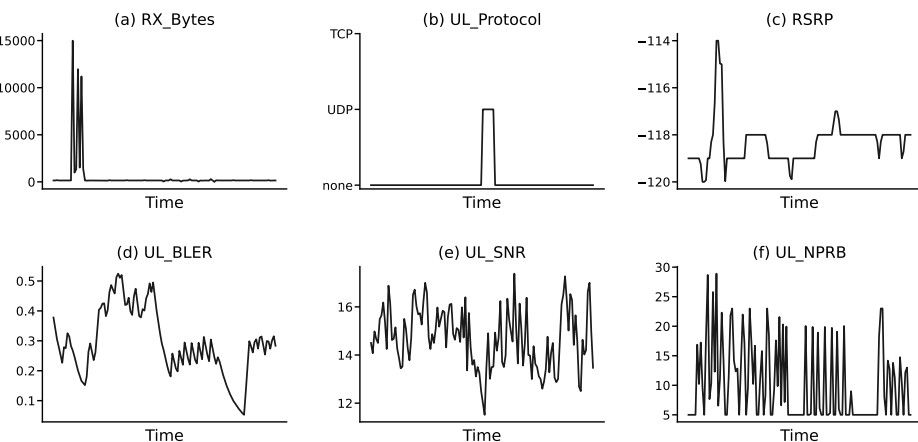

Figure 12: Randomly sampled sequence from `TelecomTS`.

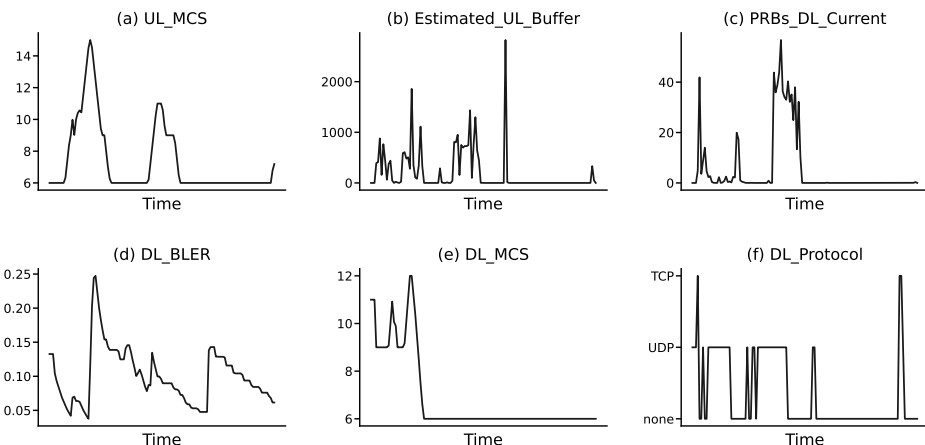

Figure 13: Randomly sampled sequence from `TelecomTS`.

# B DATA COLLECTION DETAILS

## B.1 5G NETWORK

**Overview**. To facilitate the collection of fine-grained temporal and cross-layer network KPIs from a fully operational 5G system, we implemented a standalone 5G network deployed in a controlled lab environment capable of supporting real over-the-air transmissions and enabling diverse, repeatable experimental configurations. The network consists of a single monolithic base station, connected to a software-defined radio (SDR) with a radio unit (RU) for low-level physical layer processing and signal transmission. The SDR interfaces with the gNB via a dedicated 10 Gbps Ethernet fronthaul link. Additionally, the gNB connects to the 5G core network instance over standard N2/N3 interfaces through a separate 10 Gbps Ethernet backhaul link, enabling full end-to-end standalone operation. A visual overview of the network deployment is provided in Fig. 2, where (a) shows the mobile devices and the RU, and (b) illustrates the server-side infrastructure hosting the core network and baseband functions. While the system supports multi-band operation, all experiments in this work were conducted in the n78 TDD band, using a 38.16 MHz channel bandwidth centered at 3.319 GHz.

The gNB and core network were implemented using the latest release of the open-source OpenAirInterface (OAI) software stack OpenAirInterface Software Alliance (2024). The baseband processing stack of the gNB, including the PHY, MAC, RLC, PDCP, and RRC layers, was deployed on a high-performance server equipped with an AMD Ryzen Threadripper PRO CPU (4.4 GHz, 24 cores)

and 128 GB of RAM. The 5G core network included all standard functional entities defined by 3GPP, including the access and mobility management function (AMF), session management function (SMF), user plane function (UPF), authentication server function (AUSF), network repository function (NRF), unified data repository (UDR), and unified data management (UDM). These components were deployed as containerized services within a high-availability Kubernetes cluster, hosted on a separate high-performance server with the same hardware specifications as the gNB host.

The RU was realized using a USRP N300 SDR, configured with two UBX daughterboards, each supporting up to 100MHz of instantaneous bandwidth per channel. To enhance directional transmission and reception, we utilized a 2×2 beamforming configuration provided by OAI. Finally, Google Pixel 6 and 7 smartphones, provisioned with programmable 5G SIM cards, served as the User Equipment (UE) throughout all experiments.

**Adversarial Environment Setup**. To further enhance the experimental environment and enable data collection under adversarial conditions, a suite of malicious jammers was implemented and integrated into the network. As shown in Fig. 2(c), a USRP X310 SDR was utilized to synthesize controlled over-the-air jamming signals using GNU Radio software, with the jammer strategically positioned at varying locations relative to the RU to emulate diverse radio link impairments. To introduce flexibility and realism into the adversarial environment, multiple jamming configurations were implemented, allowing dynamic control over transmission gain, occupied bandwidth, and jammer activity patterns. Throughout the campaign, three types of jamming attacks were generated: single-tone jamming (continuous narrowband interference at a specific frequency), pulsed jamming (intermittent bursts of narrowband interference), and wideband noise jamming (broad-spectrum interference across a wide frequency range). These jamming signals were transmitted over the air with the objective of disrupting the RU–UE communication link during the data collection process and observing the resulting impact across multiple KPIs, including signal quality, throughput, and error rates.

**Network Performance Tuning and Optimization**. To support long-duration experimentation and ensure reliable KPI collection with real-time granularity, several low-level software and hardware optimizations were required to maintain stable end-to-end network performance. During early operation, we observed recurring instability during measurements, often resulting in intermittent UE disconnections and incomplete KPI traces. This instability was primarily attributed to three factors: (i) the limited transmit power of the RU, which reduced link robustness during sustained over-the-air operation; (ii) processing bottlenecks on the gNB baseband stack, where high-rate IQ samples were occasionally delayed or dropped; and (iii) external in-band interference, which intermittently affected reception quality in the n78 band.

To address these challenges, we introduced a set of system-level optimizations targeting both the server running the baseband processing functions and the SDR. On the baseband server, we disabled hyper-threading to eliminate core contention, deployed a low-latency Linux kernel to reduce scheduling delays, disabled kernel page table isolation to mitigate Spectre-related overhead, and set the CPU governor to performance mode to maintain maximum CPU performance by preventing frequency scaling and disabling energy-saving states. On the RU side, the fronthaul link was carefully tuned to ensure deterministic and lossless IQ sample delivery. Jumbo frames with a Maximum Transmission Unit (MTU) of 9000 bytes were enabled to reduce packetization overhead, and both kernel socket buffers and Ethernet ring buffers were enlarged to accommodate high-throughput traffic without introducing jitter or packet drops. Finally, to minimize the impact of external interference, the operating frequency within the n78 band was selected based on in-band noise measurements, allowing us to identify and utilize the cleanest available sub-band for over-the-air transmission.

## B.2 DATA COLLECTION AND PREPARATION

**Network Zoning for Controlled Experiments.** To systematically capture KPI variations under diverse radio conditions, the network was deployed in a controlled lab environment covering approximately 70 m². The space was partitioned into three spatial zones—Zone A, Zone B, and Zone C—based on the distance between the UE and the RU. This zoning strategy enabled controlled experimentation across distinct wireless conditions, facilitating structured data collection for downstream analysis. Zones closer to the RU correspond to stronger signal conditions with minimal interference, while more distant zones experience weaker signals due to increased distance and potential obstacles.

**Zone A** includes all locations within a 3-meter radius of the RU, representing scenarios with strong signal strength, low path loss, and minimal fading.

**Zone B** spans distances between 3 and 6 meters, emulating moderate signal quality with potential variations due to partial obstruction or environmental reflections.

**Zone C** comprises all areas beyond 6 meters, corresponding to weak-signal conditions with increased attenuation, and a higher likelihood of radio link degradation.

A visual layout of the lab environment and spatial zoning configuration is shown in Fig. 14, illustrating the relative position of the RU and the boundaries of each zone.

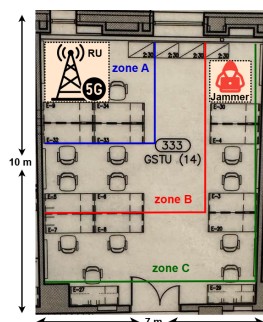

Figure 14: Spatial partitioning of the environment into 3 zones.

**Application-Level Traffic and Interference Scenarios.** To capture network behavior under representative real-world conditions, we selected a suite of application-layer scenarios encompassing both typical user behavior and adverse operational contexts. Experiments were conducted under two UE mobility profiles: (i) a static profile, where the UE remained stationary, and (ii) a mobile profile, where the UE moved at a constant pedestrian speed of 5 km/h to emulate realistic urban mobility in the lab.

The selected applications reflect common mobile usage patterns while imposing varied demands on different layers of the network protocol stack. Specifically, in the mobile device we run a set of typical mobile applications during data collection, including buffered video streaming via YouTube, live video streaming via Twitch, and large file downloads over HTTP. In addition, to examine system performance under resource contention, we introduced a controlled congestion scenario by connecting a second UE executing concurrent download tasks, thereby increasing cell load during the data collection phase.

A detailed breakdown of the data collection observations for each traffic type, mobility pattern, and zone is presented in Table 7, capturing the spatiotemporal scope of the experimental campaign across all operating conditions.

| Category | Condition | | Activity | Zones | Observations |
|---|---|---|---|---|---|
| Normal | Static | No congestion | YouTube | A, B, C | 100k/zone |
| | | | Twitch | A, B, C | 100k/zone |
| | | | File | A, B, C | 100k/zone |
| | | Congestion | YouTube | A, B, C | 10k/zone |
| | | | Twitch | A, B, C | 10k/zone |
| | | | File | A, B, C | 10k/zone |
| | Motion | | YouTube | n/a | 10k |
| | | | Twitch | n/a | 10k |
| | | | File | n/a | 10k |
| Anomalous | Jamming | | YouTube | A | 10k |
| | | | Twitch | A | 10k |
| | | | File | A | 10k |
| | Synthetic | | YouTube | A, B, C | 10k/zone |
| | | | Twitch | A, B, C | 10k/zone |
| | | | File | A, B, C | 10k/zone |

Table 7: Breakdown of total data collection sample counts across all zones and experimental conditions.

To study network behavior under adversarial conditions, we conducted controlled data collection sessions with active jamming during live application traffic. In each session, the UE maintained continuous traffic flows while exposed to over-the-air interference from a co-located jammer, allowing

us to observe both control- and data-plane KPIs under degraded radio conditions. The jammer remained stationary throughout the experiments, with its placement illustrated in Fig. 14. We employed multiple jamming patterns, including wideband noise covering the full n78 band, single-tone, and pulse-based interference. The jamming signal followed a periodic pattern, alternating between 2 seconds of activity and 8 seconds of silence. To ensure effective disruption of the RU–UE link, the jammer's transmit gain was set to 25 dBi. Representative spectrograms showcasing benign and jammed scenarios are presented in Fig. 15. These include samples of wideband noise and pulsed interference patterns used during the experiments.

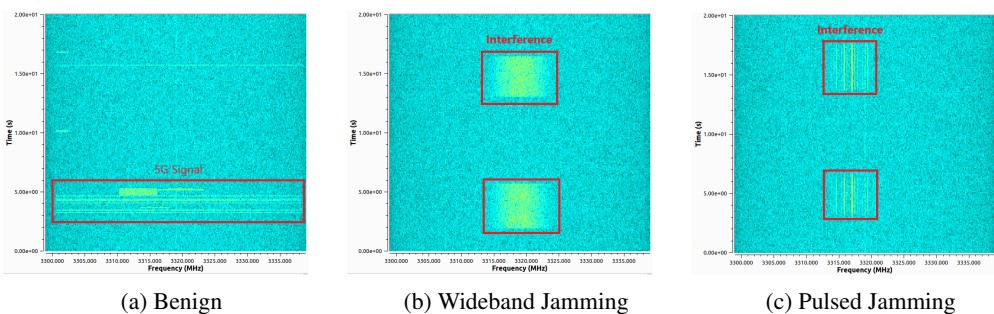

(a) Benign        (b) Wideband Jamming        (c) Pulsed Jamming

Figure 15: Spectrograms illustrating benign and adversarial interference patterns during collection.

**Data Collection, Filtering, and Synchronization**. For each measurement scenario, a single mobile device was connected to the network and actively engaged in the designated traffic session for a continuous duration of four hours. During each session, data was collected at both the link and the network layer to enable detailed analysis of network behavior. To isolate relevant traffic, all control plane signaling was excluded from the dataset. In the user plane, only transport-layer headers (i.e., TCP and UDP) were retained, while payload data was discarded to reduce storage overhead. Packet-level information was captured using Wireshark on the core network, enabling inspection of traffic characteristics and flow-level behavior, followed by IP filtering to isolate the target device.

Due to independent timestamping mechanisms between the link layer logging modules and the packet capture software, a temporal misalignment existed across the two data sources of the order of two seconds. To address this, we found the time offset that best matched the transmitted byte counts (from the Physical layer) with the downlink packet counts (from the network trace), which are expected to be highly correlated, and applied it to all PHY-layer KPIs to synchronize them with the network-layer trace. For this, we used a histogram-based matching technique: for each 300 ms window of the transmitted byte series, we computed the KL divergence against 300 ms windows of the packet count series, slid with a 30 ms stride. The best-matching offset for each window was recorded, and the mode of these offsets was selected as the final alignment correction.

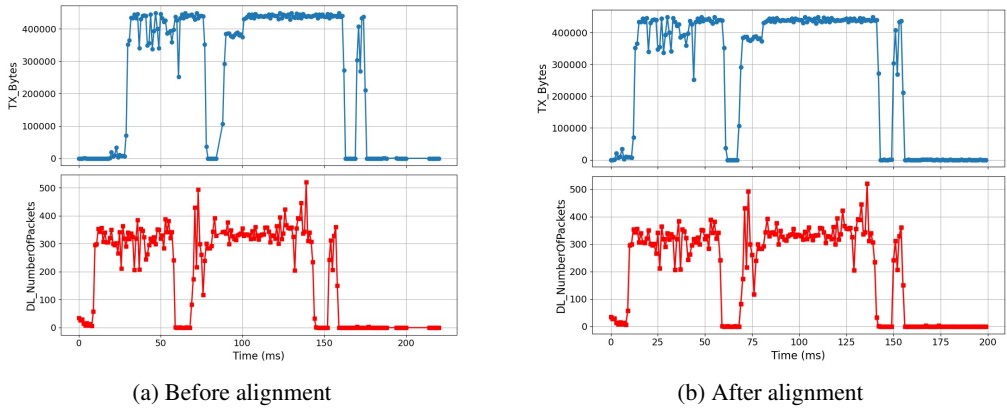

(a) Before alignment               (b) After alignment

Figure 16: Number of packets (top) and number of transmitted bytes (bottom) before (a) and after (b) alignment.

---

**Algorithm 1** End-to-End Data Collection Procedure

---

1: **Initialize Core Network**
2:   Deploy containerized 5G Core components as pods in the Kubernetes Cluster
3:   Verify inter-component connectivity between the pods
4: **Configure Radio Unit**
5:   Set sampling rate, transmit gain, and center frequency
6:   Ensure proper synchronization and signal lock
7: **Activate gNB**
8:   Launch baseband processing stack
9:   Establish registration and connection to the Core network
10: **Start Data Logging Modules**
11:   Activate KPI loggers on gNB and UE
12:   Start user plane packet capture on the Core network
13: **Connect Mobile Device**
14:   Power on mobile device and disable airplane mode
15:   Attach device to the network and establish user session
16:   Verify IP configuration and data-plane reachability
17: **Run Traffic Session**
18:   Generate traffic via selected application (YouTube, Twitch, File Downloading)
19:   Maintain session for the experiment duration
20: **Postprocessing**
21:   Filter control-plane traffic, discard payload data, and retain only packets associated with the
      target device IP
22:   Parse KPIs and packet logs
23:   Apply timestamp alignment (e.g., histogram-based matching)
24:   Export synchronized dataset for downstream analysis

---

To ensure consistency and repeatability across experiments, we followed a structured procedure that orchestrated each stage of the data collection pipeline—from system initialization to postprocessing. The detailed steps of this end-to-end process are outlined in Algorithm 1, which captures the sequence of operations for initializing the 5G network, configuring devices, capturing KPIs and traffic, and exporting the synchronized dataset for analysis.

**Overview of Collected KPIs**. To enable fine-grained monitoring of wireless performance across all protocol layers, our dataset includes a rich set of KPIs captured from both the base station and the mobile device. These KPIs span the physical (PHY), medium access control (MAC), and network layers, providing a multi-dimensional view of network behavior under varying radio, mobility, and interference conditions. The metrics include signal quality indicators, resource allocation statistics, error rates, transport protocol usage, and traffic volume, allowing detailed analysis of both control and data plane dynamics. The tables below summarize each collected KPI, including a brief description of each for reference.

---

**RSRP (Reference Signal Received Power)**

**Layer:** PHY     **Reported by:** UE     **Type:** Numerical (float)     **Range:** [-140, -45]
Measures the received signal strength from the base station's reference signals. Reflects path loss and coverage quality.

---

**UL_SNR (Uplink Signal-to-Noise Ratio)**

**Layer:** PHY     **Reported by:** UE     **Type:** Numerical (float)     **Range:** [-3.5, 60]
Indicates the uplink signal quality at the receiver. A higher SNR corresponds to better link reliability.

---

### DL_BLER / UL_BLER (Downlink / Uplink - Block Error Rate)

**Layer:** MAC      **Reported by:** gNB      **Type:** Numerical (float)      **Range**: [0, 1]
Fraction of erroneous transport blocks over total transmitted blocks in downlink/uplink. High BLER signals poor radio conditions.

### DL_MCS / UL_MCS (Downlink / Uplink - Modulation and Coding Scheme)

**Layer:** MAC      **Reported by:** gNB      **Type:** Numerical (float)      **Range**: [0, 27]
Represents the average modulation and coding level selected for a given link. Higher values indicate more aggressive transmission schemes.

### UL_NPRB (Allocated Uplink Physical Resource Blocks)

**Layer:** MAC      **Reported by:** gNB      **Type:** Numerical (int)      **Range**: [0, 105]
Number of Physical Resource Blocks assigned to the UE for uplink transmission during a Transmission Time Interval.

### Estimated_UL_Buffer

**Layer:** MAC      **Reported by:** gNB      **Type:** Numerical (int)      **Range**: [0, 250k]
Estimation of buffered uplink data at the UE as reported to the gNB via Buffer Status Reports.

### PRBs_DL_Current / PRBs_UL_Current (Downlink / Uplink - Physical Resource Blocks)

**Layer:** MAC      **Reported by:** gNB      **Type:** Numerical (float)      **Range**: [0, 105]
Number of Physical Resource Blocks currently allocated to the UE in the downlink/uplink direction in a given Transmission Time Interval.

### PRB_Utilization_DL / PRB_Utilization_UL (Downlink / Uplink - Physical Resource Block Utilization Ratio)

**Layer:** MAC      **Reported by:** gNB      **Type:** Numerical (float)      **Range**: [0, 100]
Percentage of total Physical Resource Blocks utilized by the UE in downlink/uplink over time, indicating traffic load and resource usage.

### TX_Bytes / RX_Bytes (Transmitted / Received Bytes)

**Layer:** MAC      **Reported by:** gNB      **Type:** Numerical (int)      **Range**: [0, 450M]
Total number of user-plane bytes transmitted and received, used to compute throughput and volume.

### UL_Protocol / DL_Protocol (Uplink / Downlink - Transport Protocol)

**Layer:** Network    **Reported by:** UPF    **Type:** Categorical    **Range**: {TCP, UDP, None}
Specifies the transport protocol (TCP or UDP) used in the uplink/downlink direction.

### UL_NumberOfPackets / DL_NumberOfPackets (Uplink / Downlink - Packet Count)

**Layer:** Network      **Reported by:** UPF      **Type:** Numerical (int)      **Range:** [0, 10k]
Total number of user-plane packets observed in the uplink/downlink direction.

Table 8: List of Function Types

| Function Type | Temporal | Description | Parameters |
|---|---|---|---|
| Constant Addition | ✗ | Add a fixed constant to all points | Additive Shift |
| Constant Multiplication | ✗ | Multiply all points by a fixed factor | Multiplicative Factor |
| Linear Growth | ✓ | Increase linearly | Slope |
| Exponential Growth | ✓ | Multiply data by an exponential | Growth Rate |
| Logistic Growth | ✓ | Add a logistic growth function | Growth Rate |
| Logarithmic Decay | ✓ | Multiply by decay factor | Decay Rate |
| Sinusoidal Fluctuation (additive) | ✓ | Add a sine function | Amplitude, Frequency, Shift |
| Sinusoidal Fluctuation (multiplicative) | ✓ | Multiply by a sine function | Amplitude, Frequency, Factor |

## C  ANOMALY CURATION DETAILS

**Modeling KPI Effects**. To simulate an anomaly, we apply transformations to our collected wireless data. Specifically, we alternate between sampling from two exponential distributions, one to get the anomaly inter-arrival time (the time between two anomalies) and one to get the anomaly duration. This gives us a set of timestamps $(s_i, t_i)$ of anomaly start and end time pairs. For each timestamp $(s_i, t_i)$, we will assign it some anomaly type $a_i$. Then, for the relevant affect KPIs affected by this anomaly type $(v_j^{s_i}, \ldots, v_j^{t_i})$, we apply a function to get our transformed data, $(f(v_j^{s_i}), \ldots, f(v_j^{t_i}))$.

As anomalies can have diverse effects on KPIs, as evidenced by the gathered scholarly material, we use 8 different function types listed in Table 8. Two of such function types, constant addition and multiplication, are static, while the other functions evolve with time. Next, given that many KPIs have a fixed range of possible values (e.g., UL_BLER can only be between 0 and 1), we assign boundaries to all KPIs when appropriate and truncate any values that exceed these boundaries. This often occurs with transformations such as exponential growth, where KPIs experience saturation, usually signaling a severe anomaly.

However, truncating to these "hard" boundaries can often be unrealistic, as most systems will fail before reaching theoretical saturation and not all anomalies manifest as saturations. Therefore, we set a range of soft bounds for each KPI and sample from these ranges to get our threshold. To avoid excessively aggressive soft bounds that truncate non-anomalous data, we take the threshold to be the minimum or maximum with respect to the 20th lowest or largest data point in the input time series. We choose the number 20 empirically to avoid outliers or measurement errors that may result in an issue like UL_BLER being equal to $1.06$. Furthermore, when a KPI reaches saturation, we inject noise at the saturated data points, as otherwise, we get unrealistic flat clipped values.

For specific function classes, such as sinusoidal fluctuations, linear growth, and logistic growth, we may choose to inject small noise to maintain realism for these additive effects. Additionally, a naive implementation of exponential growth leads to incredibly noisy data, often due to the presence of 0 or other small values in the data, as the data will jump between exponentially high values and near 0 within a few timestamps. For example, the transmitted bytes may be very high in general. However, for a given decisecond, it is possible that no bytes are transmitted. We remedy this by first injecting small positive noise before multiplying by the exponential factor. Furthermore, small variations in the natural noise of our data will blow up under exponential growth. Therefore, we apply kernel smoothing to get the general trend of our KPI and subtract the kernel-smoothed values from our original values to get residuals. Then, we apply exponential growth to our smoothed values and add back our residuals. This leads to realistic exponential growth behaviors with appropriate variance.

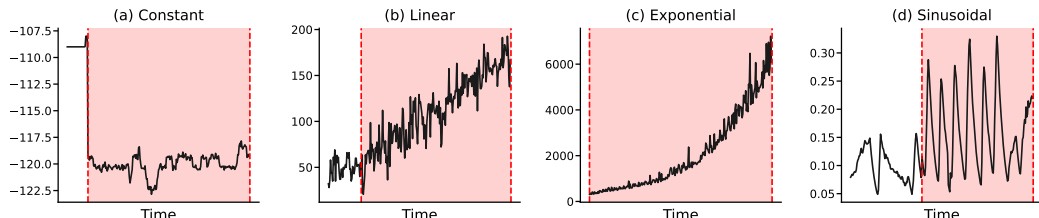

Figure 17: Examples of anomaly effects under varying function types.

Table 9: List of Anomalies

| Anomaly | Domain | Temporal | Affected KPIs |
|---|---|---|---|
| Antenna Failure | Hardware | ✓ | 13 |
| Buffer Overflow | Software/Infrastructure | ✓ | 10 |
| Co-Channel Interference (Mild) | Infrastructure | ✗ | 9 |
| Co-Channel Interference (Severe) | Infrastructure | ✗ | 11 |
| Doppler Shift | Environment | ✓ | 7 |
| Faulty Handover Algorithm (Frequent) | Software | ✓ | 9 |
| Faulty RF Filters | Hardware | ✓ | 9 |
| High Network Congestion (Static) | Usage | ✗ | 10 |
| High Network Congestion (Temporal) | Usage | ✓ | 10 |
| Resource Allocation Bugs | Software | ✓ | 9 |

**Anomaly Curation**. We carefully select a list of 10 representative anomalies listed in Table 9 that span diverse effects on KPIs. These anomalies can be classified into one of five types of wireless anomalies: hardware failure, software issues, infrastructure issues, environmental interference, and anomalous usage. Three of our anomalies are static, meaning that all KPIs are affected statically, reflecting sudden onset anomalies. The remaining anomalies are temporal and represent anomalies that gradually build up. For every anomaly, we use scholarly material to select a list of KPIs that would be affected under this anomaly and match each KPI with a function class/KPI effect mentioned in the previous section. Most importantly, to accurately simulate anomalies, we carefully pick parameters for these functions classes to match the anomaly. Given an affected KPI of an anomaly, we use GPT-4.1 to generate a range of feasible values for each parameter. Using a range allows us to model the stochasticity of real-world anomalies. Then, at generation time, we will uniformly sample within this range to determine the transformation applied to our data. We repeatedly verify these parameters using human feedback until we have satisfactory results.

**Troubleshooting Ticket**. We provide GPT-4.1 with the prompt found in Appendix D to generate a troubleshooting ticket each time we simulate an anomaly in our wireless data. The anomaly impact variable inputs a fixed textual description for each anomaly that lists how every affected metric changes under the anomaly. The alarm time and resolution time are optional inputs that match the start and end time of the corresponding anomaly. We use a human-in-the-loop process to ensure quality for our tickets.

**Anomaly Dataset.** To obtain a final anomaly dataset, we slice all anomalous time series sequences into sequences of length 128 with a stride of 32. We remove all samples that no longer contain anomalous data points. We also remove all samples containing two different types of anomalies to simplify downstream tasks and to avoid unrealistic anomaly density, as recommended by Wu & Keogh (2023). Finally, we add relevant metadata such as alarm and resolution time, affected metrics, and an indicator array for anomalous data points.

## D  PROMPTS

**Troubleshooting Ticket.** This prompt is used to generate multimodal anomaly detection and root cause analysis simultaneously with the simulated time series anomaly data in Appendix C. For every instance of an anomaly, we record its type (for instance "Antenna Failure") as well as its impact on metrics. The latter is a paragraph that describes how each affected KPI is transformed under the anomaly and its corresponding function type. Optionally, we can include the alarm and resolution time. We prompt GPT-4.1 to perform a root cause analysis and generate a hypothetical solution to the simulated anomaly. Such troubleshooting tickets can hopefully endow multimodal anomaly detection models with anomaly analysis and resolution insights.

> **Troubleshooting Ticket Generation Prompt**
>
> Generate a concise troubleshooting summary for a wireless network anomaly.
>
> **Context:**
>
> - **Anomaly Type:** `anomaly_type`
> - **Alarm Time:** `anomaly_time`   **[Optional]**
> - **Resolution Time:** `resolved_time`   **[Optional]**
> - **Anomaly Impact:** `anomaly_impact`
>
> **Format the response as follows (DO NOT add extra explanations):**
> **Diagnose Summary:**
>
> - **Issue:** [Briefly describe the detected anomaly.]
> - **Symptoms:** [Summarize affected metrics and key changes.]
> - **Root Cause:** [State the most likely cause.]
> - **Resolution:** [Summarize the main actions taken to fix it.]

**Anomaly Experiment Prompts.** The following prompts are used to assess a model's anomaly detection and analysis capabilities. For each prompt, we provide a time series sample as well as the alarm and resolution times. Detailed instructions and context are given, and we optionally provide additional context on wireless data or anomaly descriptions to aid the model. Finally, we prompt the model to output a strictly formatted conclusion block that allows for regular expression parsing.

**Anomaly Detection Prompt**

You are an AI assistant tasked with analyzing time series data for anomalies in a wireless network. You will be provided with a time series dataset containing various metrics and a specific time range to analyze. The time series is sampled every 0.1 seconds (i.e. timestamps are a decisecond apart), and contains a total of $n$ time steps. Your goal is to detect any anomalies within this range and identify the timestamps where they occur.

**[Optional, if `context=True`]**
Note: Wireless network data is naturally **noisy and erratic**, even under normal conditions. Sporadic spikes, sharp drops, or momentary fluctuations can appear **without indicating any true anomaly**. This sequence is only $\ell$ seconds long, so be especially cautious in interpreting short-term changes as significant. Only mark something as anomalous if there is **clear and sustained evidence** of abnormal behavior across multiple metrics.

First, review the time series data provided from `start_time` to `end_time`:

```
metric_1: v_1^1 v_2^1 ... v_n^1
metric_2: v_1^2 v_2^2 ... v_n^2
more metrics ...
```

**[Optional, if `context=True`]**
To detect anomalies, follow these steps:

1. Begin by scanning the time series for any unusual behavior: sharp spikes or drops, sustained deviations, or values inconsistent with the expected range.

2. Consider inter-metric relationships — for example, whether high buffer utilization coincides with low throughput or high BLER.

3. All anomalies occur at the same timestamp range, so you should identify a single set of timestamps for the anomaly event and attribute affected metrics to that period.

Summarize your conclusion as follows:

```
<conclusion>
Anomaly Detected:  [Yes/No]
[If yes, include the following strictly formatted line:]
Anomaly Timestamps:  [(start_time1, end_time1),
(start_time2,
end_time2), ...]
</conclusion>
```

Only base your analysis on the provided time range. If no anomaly is detected, write:

```
<conclusion>
Anomaly Detected:  No
</conclusion>
```

Do not include additional comments or summaries outside this format.

---

**Anomaly Boundary Prompt**

You are an AI assistant tasked with analyzing time series data for anomalies in a wireless network. You will be provided with a time series dataset containing various metrics and a specific time range to analyze. The time series is sampled every 0.1 seconds (i.e. timestamps are a decisecond apart), and contains a total of $n$ time steps. Your goal is to identify a **single contiguous time interval** during which an anomaly occurs. There is exactly one anomaly in the data, and it may span the entire sequence or just a sub-segment.

First, review the time series data provided from `start_time` to `end_time`:

```
metric_1: v_1^1 v_2^1 ... v_n^1
metric_2: v_1^2 v_2^2 ... v_n^2
more metrics ...
```

Summarize your conclusion as follows:

```
<conclusion>
Anomaly Timestamps:  (YYYY-MM-DD HH:MM:SS.sss, YYYY-MM-DD
HH:MM:SS.sss)
</conclusion>
```

Do not include any additional commentary or explanation outside the specified format. Respond with *only* the `<conclusion>` block and nothing else.

---

**Root Cause Analysis Prompt**

You are an AI assistant tasked with diagnosing a known anomaly in wireless network time series data. You will be provided with a short time series segment sampled every 0.1 seconds, covering $\ell$ seconds and $n$ time steps. This sequence ranges from `start_time` to `end_time` and **is confirmed to contain an anomaly**.

The anomaly is known to be **one of the following**, and each is equally likely to occur in this dataset. **Do not assume any anomaly is more common or more likely than another.**

Your task is to identify the most plausible anomaly type **from the following list:** `anomaly_list`

Please analyze the metrics below and select the **single most likely anomaly**.

**[Optional, if `descriptions=True`]**
Here is a summary on how the provided anomalies generally behave: `[Anomaly descriptions]`

Here is the time series data:

```
metric_1: v_1^1 v_2^1 ... v_n^1
metric_2: v_1^2 v_2^2 ... v_n^2
more metrics ...
```

Summarize your conclusions as follows:

```
<conclusion>
Anomaly Type:  [One exact string from the predefined
anomaly
list.]
</conclusion>
```

Do not include any additional commentary or explanation outside the specified format. Respond with *only* the `<conclusion>` block and nothing else.

---

**Time Series QA Prompts**. These prompts are used to assess a model's time series analysis capabilities. For each prompt, we provide a single KPI from a sample and ask the model to perform elementary statistical analysis such as detecting the average value, variance, etc. Oftentimes, models will output reasoning steps, so we include warnings to discourage such behavior, which has significantly helped with regular expression parsing.

> **Mean Detection Prompt**
>
> Consider the following list of numbers representing a time series: $v_1, v_2, \ldots, v_n$. Some values may be missing (NaN). What is the average `channel` value of this series, ignoring NaNs? Respond with only a single float rounded to 2 decimal places — no other text or numbers. Please DO NOT include any other analysis or explanations.

> **Variance Detection Prompt**
>
> Consider the following list of numbers representing a time series: $v_1, v_2, \ldots, v_n$. Some values may be missing (NaN). What is the variance of `channel` for this series, ignoring NaNs? Respond with only a single float rounded to 2 decimal places — no other text or numbers. Please DO NOT include any other analysis or explanations.

> **Periodicity Detection Prompt**
>
> Consider the following series: $v_1, v_2, \ldots, v_n$. Please investigate whether the series exhibits strong periodicity, ignoring any NaN values. If it does, respond with an integer value representing approximately how often strong periods occur in the series. If there is no evidence of strong periodicity, respond with the sequence length $n$. Do not include any other numbers in your response, whether in the form of intermediate calculations or steps. Remember you MUST return an INTEGER value or $n$. Please DO NOT include any other analysis or explanations.

> **Trend Detection Prompt**
>
> Consider the following series: $v_1, v_2, \ldots, v_n$. Please describe the average trend of the series, ignoring any NaN values. If the series is decreasing on average, respond with a value of -1. If it is increasing, respond with a value of 1. If there doesn't appear to be a strong trend in any direction, please respond with a value of 0. Note that wireless data can be noisy, so look at global changes to determine trend. Do not include any other numbers in your response, whether in the form of intermediate calculations or steps. ONLY RESPOND WITH -1, 0, or 1. Do NOT include any other analysis or explanations.

**Network QA Prompts**. These prompts are used to assess a model's network understanding capabilities. For each prompt, we provide the KPIs from a sample and ask the model to provide an answer to the question at hand.

**Network QA Prompt**

You are an AI assistant tasked with analyzing time series data for a wireless network. You will be provided with a time series dataset containing various metrics to analyze. The time series is sampled every 0.1 seconds.

Your goal is to answer the questions about the user's activity, location, network congestion, jammer presence, and motion status based on the provided time series data only.

The possible activities are: YouTube, Large file download, and Twitch. The possible zones are: Zone A (closest to the gNB), Zone B (middle), and Zone C (furthest). The possible congestion status is: Yes or No. The possible motion status is: Yes or No. The possible jammer presence is: Yes or No.

Time range : $\{ts\_range[0]\}$ to $\{ts\_range[-1]\}$
$\{metric_1\}$ : $\{values_1\}$
$$\vdots$$
$\{metric_n\}$ : $\{values_n\}$

Now, answer the following questions:
Q1. What activity was the user engaged in?
Q2. Where was the user located?
Q3. Was the network congested?
Q4. Was the user in motion?
Q5. Was there a jammer present?

Do not include any reasoning, explanation, or commentary. You must return only the final answer using the format shown below, exactly as specified.

Respond with:
```
<activity>[your answer here]</activity>
<zone>[your answer here]</zone>
<congestion>[your answer here]</congestion>
<motion>[your answer here]</motion>
<jammer>[your answer here]</jammer>
```

## E TELECOMTS: AN EXAMPLE

**Root Cause Analysis Prompt**

```json
{
    "start_time": "2025-07-07 00:07:21.600",
    "end_time": "2025-07-07 00:07:34.300",
    "sampling_rate_hz": 10,
    "KPIs": {
        "keys": [
            "RSRP", "DL_BLER", "DL_MCS", "UL_BLER", "UL_MCS",
            "UL_NPRB", "UL_SNR", "TX_Bytes", "RX_Bytes",
            ...
        ],
        "values": [
            [-106.0, -106.0, -106.0, ... ], [0.00132, ... ],
            ...
        ]
    },
    "anomalies": {
        "exists": true,
        "type": ["High Network Congestion (Gradual Buildup)"],
        "anomaly_duration": [{"start": 0, "end": 127}],
        "affected_kpis": ["UL_BLER", "TX_Bytes", ... ],
        "troubleshooting_tickets": ["High Network Congestion",
            "**Diagnose Summary:**\n- **Issue:** ... "]
    },
    "statistics": {
        "RSRP": {
            "mean": -106.0,
            "variance": 0.0,
            "trend": 0,
            "periodicity": 1
        },
         ...
    },
    "labels": {
        "zone": "B",
        "application": "Youtube",
        "mobility": "No",
        "congestion": "No",
        "anomaly_present": "Yes"
    },
    "QnA": {
        "network": [
            {"q": "Can we classify the user as moving?",
             "a": "The session involved a static user."},
             ...

        ],
        "timeseries": [
            {"q": "What is the var of RX_Bytes?", "a": 286.3},
            {"q": "What is the avg value of RSRP?","a": -106},
             ...
        ]
    },
    "description": "The radio link shows a steady downlink..."
}
```

## F  TRAINING DETAILS

We train our models on `TelecomTS` using an 80–20 split between training and test data. To avoid label imbalance and potential bias, we ensure that the training subset is balanced across labels, both for anomaly detection and root cause analysis tasks. For foundation models, only the classification or regression head is trained, while the backbone remains frozen. Optimization is performed using the Adam optimizer with a learning rate of 0.0001, a batch size of 64, and for 10 epochs. The training objective depends on the task: cross-entropy loss is used for classification, while mean squared error (MSE) is used for forecasting.

