# OpenReview forum: "TelecomTS: Observability Dataset for Multi-Modal Time-Series and Language Analysis"
_ICLR.cc/2026/Conference — Submitted to ICLR 2026_

### Official Review · Reviewer_uXMG · 2025-10-18

**Soundness:** 4
**Presentation:** 4
**Contribution:** 1
**Rating:** 2
**Confidence:** 5

**Summary:**

A real-world observability dataset in time series domain, de-anonymized and without normalization.

**Strengths:**

1. Built on a real-world physical system.
2. De-anonymized.
3. No normalization, preserving the original data scale.
4. High resolution.
5. Synthetic anomalies appear reasonable, supported by (i) a literature review and (ii) human evaluation;however, please note **this is not a human-in-the-loop verification process, it is an open procedure without feedback**.
6. Good insight: in certain threshold-triggered systems, the erratic, zero-inflated nature of these series makes forecasting less critical; anomaly detection and multi-step reasoning for downstream decision-making are more important.

**Weaknesses:**

1. The paper feels incomplete. It claims to include a Q&A dataset, but there appear to be only 64 time-series Q&A and 5 network Q&A samples (as shown in Table 1). I may be misunderstanding, but **if Network QA indeed has only 5 samples, that is far too few;** results based on such a small sample are not statistically meaningful.
2. There is an apparent inconsistency: the paper argues that the erratic, zero-inflated nature of these series makes forecasting less critical, yet the experiments still focus on forecasting (Section 4.4). This conflicts with the Introduction; **this work should focus on downstream control for network congestion and anomaly fixing rather than prediction.**
3. In Section 4.1, all models perform poorly, which raises **concerns about data quality**. If the dataset contains substantial noise, you should place greater emphasis on preprocessing and provide insights into data quality and preprocessing. ***A dataset alone, without meaningful insights, is engineering rather than research.***
4. A similar issue appears in Section 4.2. The anomaly duration analysis essentially reduces task difficulty; I do not think this setting is necessary. Instead, focus more on preprocessing to **ensure data quality**.
5. Root cause analysis should include more details.

Overall, the paper is ***incomplete and lacks useful insights***; I mainly see engineering effort rather than research contributions.

**Questions:**

Please refer to the weaknesses.

---

> ### Author Response · Authors · 2025-11-21
> **Rebuttal Reviewer uXMG**
>
> We thank reviewer **uXMG** for the review, and we would like to address several **fundamental** misunderstandings by the reviewer and reiterate the research contributions and insights of our paper. We provide a point-by-point clarification below.
>
> - **Anomaly curation is not human-in-the-loop, but open procedure without feedback (S5):** We believe there is a misunderstanding here. To clarify our process for curating synthetic anomalies, we iteratively refined **both** the anomaly modeling approaches and the troubleshooting tickets using expert feedback. An open procedure without feedback would imply that expert input does not influence our synthetic anomalies, which is **not the case**. Instead, our process is cyclical: experts review the anomalies and their corresponding tickets, we update the modeling accordingly, and the cycle repeats. This iterative refinement is how we ensured that our synthetic anomalies align with how such anomalies would manifest in real-world scenarios.
> - **Only 64 time-series Q&A and 5 network Q&A samples (W1):** For clarification, we have 69 **types** of questions for **each** time-series sample, as seen in Table 1. This gives us a total of  69*32,000=2208000 QAs, which is a large set of QAs involved. To avoid future confusion, we will edit Table 1 to include this number, not just the question type.
> - **The paper should not include forecasting, as it argues it is less critical (W2):** We precisely include the forecasting task to **justify** our point in the introduction. Our analysis and results show how zero-inflated data lead to deceptively low MAE and RMSE scores, in Table 5, despite failures in forecasting, as illustrated in Figure 7.  Furthermore, forecasting is the standard evaluation used across the time-series community; hence, we include it to help ground viewers from this community. It is also worth noting that our benchmark extends far beyond forecasting, with tasks such as anomaly detection, root cause analysis, and Q&A, among others.
> - **All models perform poorly in 4.1 and 4.2, raising concerns about data quality (W3, W4).** This statement is incorrect, as we observe adequate and strong performances across all three anomaly tasks in Sections 4.1 and 4.2 from multiple models. For section 4.1, **Mantis** achieves an F1 score of **0.711** on anomaly detection. Across other tasks, we observe **Toto** achieving a high F1 score of **0.921** for anomaly duration and a very high accuracy of **0.848** for root cause analysis. Mantis also achieves 0.893 F1 on anomaly duration. This indicates that errors have more to do with model performance and **not** noise from data. Additionally, poor model performance alone is not sufficient evidence of poor data quality. As we show in Section 4.1, LLMs performed poorly on the tasks, but this is due to their limited ability in time-series reasoning. Nevertheless, they remain an important baseline given their widespread use and zero-shot applicability, which is why we included them.
> - **Anomaly duration essentially reduces task difficulty and isn’t necessary (W4):** We disagree with this statement and believe that anomaly duration is necessary. First, anomaly detection and duration are two **fundamentally different** tasks. To clarify, the anomaly detection simply detects whether there **is** an anomaly or not. Anomaly duration assumes we already have an anomaly, and we must identify which **precise** timestamps the anomaly occurs at. This requires understanding short time-series horizons to localize the anomaly, while detection could rely on more global behavior. Thus, we test a unique aspect of the model's capability through this task. Furthermore, anomaly duration is a critical task in real-world scenarios. For example, in network performance analysis, knowing the duration of an anomaly is extremely valuable because it pinpoints where the issue occurs, how long it lasts, and thus deduces how severe it is.
> - **Details on Root Cause Analysis (W5)**: Essentially, we provide each model with a full time-series sample containing an anomaly and ask it to classify the specific anomaly type. Time-series models are trained with a classification head, while LLMs are given a list of anomaly types along with descriptions of how they typically manifest to support their reasoning. To further improve clarity, we will move some of the training details currently in Appendix F into the main results section on root cause analysis.
>
> With the above in mind, we emphasize that TelecomTS is the first observability dataset with explicit scale information that showcases how crucial such information is, and opens avenues for research on how best to incorporate scale into downstream tasks. It is also worth noting that this is a **dataset paper**, where the value lies both in the gaps it fills within existing datasets and in the methodological directions it enables (for example, the integration of scalar information in multimodal observability settings).

---

> > ### Comment · Reviewer_uXMG · 2025-11-26
> >
> > I acknowledge and appreciate the authors’ rebuttal. Nevertheless, the key concerns I raised above are, in my view, still insufficiently addressed, so my overall evaluation remains the same.
> >
> > Since Figure 3, Table 1, and the explanation around W5 led to misunderstanding, I believe the authors should revise and upload a new version of the paper.
> >
> > Regarding W1, the authors clarify that there are in total 69 * 32,000 = 2,208,000 QA pairs. However, this raises a further question: how is the quality of these QA pairs ensured? The fact that the total number can be written so neatly suggests that these QA pairs were generated in bulk, without substantial quality checking or filtering of low-quality samples, and that experiments were then run directly on these uncurated outputs.
> >
> > Regarding W2, my main point is that, given such a dataset and a manually constructed environment, the focus should be on downstream control for network congestion and anomaly fixing, rather than on forecasting alone. The authors have not directly addressed this concern.
> >
> > Regarding W3, as in W1, my primary concern remains the overall quality of the dataset.
> >
> > I still believe this work is incomplete and lacks useful insights; to be frank, I did not learn anything substantial after reading this paper. I would recommend that the authors, based on TelecomTS dataset, at least perform some model fine-tuning or model specialization, in order to ensure and demonstrate the quality and usefulness of the data.

---

> ### Author Response · Authors · 2025-11-27
> **Second Rebuttal Reviewer uXMG**
>
> We first thank the reviewer for taking the time to respond to our rebuttal. We still believe that the reviewer’s comments contain misinterpretations that we clarify below:
>
> 1. **Regarding W1:** Our Q&As are generated from long-running experiments conducted in a fully controlled network environment. Because the environment is controlled, we have access to ground truth regarding congestion, user mobility, current network activity, etc. Given such control, we are certain about the factual accuracy and reliability of the generated Q&As. Therefore, the large volume of Q&A samples does not inherently call into question their quality. We also refer the reviewer to Appendix B, where we provide extensive details on the data collection setup, the documented metadata, and the post-processing procedures applied.
> 2. **Regarding W2:** We agree that forecasting should not be the sole focus of the paper, and we emphasize that this is **not** the case for our paper. In addition to forecasting, we have considered several other tasks, including anomaly detection, root cause analysis, anomaly duration estimation, and question-answering. Forecasting constitutes only **one** of our experiments, and its role is to illustrate the inherent difficulty of forecasting in observability domains. We note that many time-series works include multiple tasks, such as anomaly detection and forecasting, given that forecasting is a standard evaluation benchmark within the research community [1,2,3,4].
> 3. **Regarding W3:** We reiterate our previous response that several models exhibit strong performance on our dataset, and we explicitly discuss in the paper which models perform well and which do not, along with the underlying reasons (e.g., the Mantis model achieves 0.711 F1-score on the anomaly detection task). Therefore, poor performance by certain models does not imply issues in our dataset quality.
> 4. **Model Fine-tuning:** We emphasize that, aside from LLMs such as GPT-4.1, all models are fine-tuned on TelecomTS, including time-series foundation models such as Moment, Moirai, and others.
>
> Finally, we address the reviewer’s concern regarding the insights provided. We emphasize that our work is a **dataset paper** that bridges a crucial gap in current observability datasets, as it introduces the first observability dataset with explicit scale information. It also offers insights into the importance of scale incorporation in time-series modeling and highlights the research gaps that exist in the current landscape. Dataset papers are fundamentally valuable in this respect, as their contributions lie in filling gaps in existing datasets and enabling new methodological directions [5,6]. We hope this clarifies the reviewer’s concerns and makes the contributions of the paper more evident.
>
> [1] "Time Series PILE Benchmark”, Auton Lab 2024.
>
> [2] "TimesNet: Temporal 2D-Variation Modeling for General Time Series Analysis”, ICLR 2023.
>
> [3] “Peri-midFormer: Periodic Pyramid Transformer for Time Series Analysis”, NeurIPS 2024.
>
> [4] "Time-MQA: Time Series Multi-Task Question Answering with Context Enhancement”, ACL 2025.
>
> [5] “Time-MMD: Multi-Domain Multimodal Dataset for Time Series Analysis”, NeurIPS 2025.
>
> [6] “The Elephant in the Room: Towards A Reliable Time-Series Anomaly Detection Benchmark”, NeurIPS 2024.

---

### Official Review · Reviewer_aULz · 2025-10-30

**Soundness:** 3
**Presentation:** 3
**Contribution:** 2
**Rating:** 4
**Confidence:** 3

**Summary:**

The paper proposes a telecom observability benchmark built from high-cadence (multi-KPI) device/base-station traces, including induced conditions (mobility, congestion) and a real jammer setup. The authors procedurally synthesize anomaly segments from literature-derived types and KPI-level symptom mappings, and use LLMs to generate ticket text/templates. The benchmark evaluates anomaly detection, root-cause classification, forecasting, and QA, with LLM and time-series model baselines.

**Strengths:**

- Thorough real-world data collection and setup: The authors instrument 18 KPIs at high cadence, align trace, spatial zones, vary mobility and congestion, with a real jammer. I think this setup is thorough.

- Broad task suite: The work evaluates anomaly detection, anomaly-type classification, forecasting, and QA with clear context settings and failure analyses.

- Insight on scale handling: Results highlight the importance of preserving absolute scale for this specific type of domain. Models that encode local mean/std (Mantis-style features) perform robustly.

- Principled synthetic pipeline: Anomaly types are grounded in telecom literature with KPI-symptom mappings, parameterized timings, and human verification, providing controllable coverage beyond rare real incidents.

**Weaknesses:**

1. QA tasks emphasize statistic extraction rather than reasoning.

   Labels for several QA tasks are deterministic outputs of simple statistics (means/variances, FFT periodicity, linear trend), so tasks can be solved by lightweight scripts or classifiers. Even for the root cause analysis, it can be framed as a 1 step classification tasks. I don't think these tasks strongly probe multi-hop or cross-channel reasoning.

2. Ambiguity between LLM performance and LLM usage.

   The paper reports that GPT-style models underperform on anomaly detection, yet LLMs are used in the pipeline (for ticket/text generation). This invites confusion unless the boundary is stated explicitly and the contribution of tickets to downstream performance is measured through ablations (see more in question 2).

3. Missing ultra-simple baselines and efficiency context.
   Strong but simple baselines such as DLinear ([arXiv:2205.13504](https://arxiv.org/abs/2205.13504)), AutoAR ([arXiv:2411.02796](https://arxiv.org/abs/2411.02796)), per-KPI linear regression ((https://openreview.net/pdf?id=wfyc8vLcq0)), and naïve last/seasonal predictors are absent as comparison models. Given recent evidence that simple models can match or beat heavier foundation models in time series, this limits interpretability; compute/latency reporting is also desired.

4. Prompting & evaluation fragility for LLMs

    For anomaly/RCA experiments, models are given optional context about KPI behaviors and asked to output a strictly formatted "conclusion block" for regex parsing. Results may be highly prompt-sensitive; mis-formatting can incur evaluation penalties unrelated to capability. (Please report prompt ablations and parsing robustness.)

**Questions:**

- How do the authors quantitatively validate real-vs-synthetic similarity (e.g., KS/MMD on KPI features, or train-on-synthetic→test-on-real and the reverse)?
- The duration/inter-arrival processes are modeled as exponential; how sensitive are results to that choice? Were alternative priors considered?
- Please clarify the boundary: LLMs generate only tickets/templates without attending to the time series right? If so, does adding tickets measurably improve multimodal performance versus a no-ticket ablation?
- An external validation of some of your key insights would help (for example, does scale handling helps on an external observability dataset as well?)

---

> ### Author Response · Authors · 2025-11-21
> **Rebuttal Reviewer aULz (1/2)**
>
> We thank **reviewer aULz** for their feedback and for highlighting our thorough data collection setup and broad task suite. We appreciate your suggestions and address your questions below:
>
> 1. **QA tasks emphasize extraction, not reasoning (W1):** Our dataset includes several types of questions. Some of them, as noted by the reviewer, involve extraction-focused tasks such as predicting trends in the time-series KPIs. However, another suite of tasks, particularly the networking-specific questions and anomaly-related tasks such as detection and root cause analysis, require more in-depth reasoning. For example, in root cause analysis, the model must reason over the time-series itself and also reason about the symptoms that each anomaly type is known to manifest before arriving at an answer. Therefore, these tasks go far beyond simple extraction of statistics from the time-series KPIs.
> 2. **Ambiguity between LLM performance and LLM usage (W2, Q3):** We clarify this ambiguity below. First, it is correct that we use LLMs to generate troubleshooting tickets. However, during ticket generation, the LLMs have access to explicit information, such as the cause of the anomaly, the anomaly’s start and end times, and its typical behavior, all grounded in established telecom literature. The LLM then synthesizes this information into troubleshooting tickets with accuracy and ease. In contrast, during evaluation, we provide the LLM only with the raw time-series data and high-level contextual knowledge about the possible anomaly types and their expected manifestations (we do not provide it with the whole troubleshooting ticket as that includes the ground truth anomaly information). The LLM must then reason over the time-series and use this textual context to infer whether an anomaly is present and, if so, determine its type. In our results, we show that current SOTA LLMs perform poorly on this task, as they struggle to jointly reason over time-series patterns and textual domain context.
> 3. **Missing ultra-simple baselines (W3):** As per your suggestion, we have included DLinear and per-KPI linear regression here. We agree this is a valuable comparison, given the success of simple baselines. For DLinear, we hyperparameter-tune through a grid search across learning rate and weight decay. For the linear regression, we grid search with the same learning rate and weight decay grid from the original paper. Our context length is 4, 8, 16, 32, 64, and we use early stopping.  We highlight, however, that these baselines can only be applied to forecasting tasks and do not support the other multimodal tasks considered in our papers.
>
>
>     | Forecasting | DLinear | per-KPI Lin Reg | Moment | Moirai | Toto | Mantis | TimesNet | Autoformer | Non-stationary Transformer | FEDformer | Informer |
>     | --- | --- | --- | --- | --- | --- | --- | --- | --- | --- | --- | --- |
>     | MAE | 0.6967 | 0.1629 | 0.5435 | 0.5160 | 0.4896 | 0.4578 | 0.1595 | 0.4584 | 0.2563 | 0.1702 | **0.1437** |
>     | RMSE | 1.0730 | 0.3953 | 0.7216 | 0.6988 | 0.6759 | 0.6037 | 0.3964 | 0.8948 | 0.5608 | 0.4080 |  **0.3586** |
> 4. **Compute/latency reporting is also desired (W3):** We originally did not include latency reporting, as API-level latency is highly variable and can be misleading because it includes factors such as network delays. That being said, compute and latency follow the expected trends: larger models, such as LLMs, require more FLOPs and exhibit higher latency, while smaller models run faster with lower compute requirements. We will include the time complexities of the architectures found in our evaluation in the camera-ready version of the paper.
> 5. **Prompt ablations and parsing robustness (W4):** To test the robustness of the results, we investigated the models’ formatting of the ‘conclusion block’ for regex parsing. We observed that the majority of LLM outputs produced the block correctly, allowing us to parse it reliably with regex. Formatting errors occurred in only about 3% of cases, and when such errors did occur, we regenerated the model’s response to ensure a properly formatted answer. Additionally, in our trials, we did not observe significant performance differences whether or not the model was explicitly prompted to structure and format its output. We believe that, given that the models we use are state-of-the-art models, such minor variations in prompts did not lead to major performance changes.

---

> > ### Author Response · Authors · 2025-11-21
> > **Rebuttal Reviewer aULz (2/2)**
> >
> > 6. **Quantitatively validate real-vs-synthetic similarity (Q1):** Ideally, we would have liked to have access to substantial real anomalies in the literature for a variety of network issues, which would allow us to quantitatively validate our synthetic anomalies. Unfortunately, such datasets do not exist. This is precisely why our proposed dataset is important, as it aims to bridge this gap, as current observability datasets in domains such as networking are limited, largely due to their proprietary nature. However, to ensure our synthetic anomalies closely mimic real ones, we took several steps, including grounding their behaviors in telecom literature (for example, if a particular anomaly is known to manifest in a specific way, our synthetic anomaly mirrors that behavior), along with expert review.
> > 7. **The duration/inter-arrival processes are modeled as exponentials (Q2):** We only consider the exponential prior because it is grounded in real-world behavior and verified in the literature [1-3]. However, our results are not sensitive to this choice. Our samples are divided uniformly with a length of 128 and a stride of 32. Therefore, changing the prior would only affect the anomaly duration distribution within each sample, while the performance on anomaly detection, root cause analysis, and other tasks would remain similar. For this reason, we chose the prior that aligns with real-world behavior as observed and verified in the literature.
> > 8. **External validation of key insights (Q4):** Observability datasets available in the literature are typically normalized and anonymized (e.g., BOOM) due to their proprietary nature. As a result, there are no external observability datasets on which we can perform external validation. However, to further assess the importance of scale, we conduct a direct ablation on Mantis using our dataset. Specifically, Mantis encodes scale information in its tokenizer through NME scalar encoders [4]. We remove these encoders so that Mantis is trained only on normalized data using standard z-normalization without any explicit scale information. We report the results below.
> >
> >
> >     | **Task \ Model Variant** | Mantis | Mantis w/o NME for scaling |
> >     | --- | --- | --- |
> >     | Anomaly Detection (Pr, Rec, F1) | (0.640, 0.800, **0.711**) | (0.362, 0.850, 0.510) |
> >     | Root-Cause (Acc) | **0.590** | 0.525 |
> >     | Anomaly Duration (Pr, Rec, F1) | (0.8734, 0.9144, **0.8934**) | (0.8033, 0.9704, 0.8790) |
> >     | Forecasting (MAE, RMSE) | (**0.4578,** **0.6037**) | (0.5703, 0.8436) |
> >
> >     As expected, we observe a decrease in performance when scalar information is removed. This highlights the importance of TelecomTS, as it provides explicit scale information and opens up avenues for exploring how to best incorporate such scale information to achieve optimal performance. We will include these results in the camera-ready version.
> >
> >
> > [1] "A Framework for the Evaluation of Network Reliability Under Periodic Demand”, IEEE Transactions on Networking.
> >
> > [2] “Probability and Life Distributions for Reliability Analysis” in *Reliability Engineering*, Wiley.
> >
> > [3] “Failure and Reliability of Electronic Materials and Devices” in *Engineering Materials Science*, Academic Press.
> >
> > [4] “NuTime: Numerically Multi-Scaled Embedding for Large-Scale Time-Series Pretraining”, TMLR 2024.

---

> > > ### Author Response · Authors · 2025-11-27
> > > **Message to Reviewer aULz**
> > >
> > > Dear Reviewer aULz,
> > >
> > > Thank you once again for your valuable time and feedback. We hope our response has satisfactorily addressed your concerns, and we would be glad to provide any further clarification if needed.
> > >
> > > Best Regards,
> > >
> > > The Authors

---

> > > > ### Comment · Reviewer_aULz · 2025-11-28
> > > >
> > > > I thank the author for the detailed response to my questions. After a careful review, I believe most of these answers address my questions earlier. Therefore I am raising my score for this paper.

---

### Official Review · Reviewer_PdGG · 2025-10-31

**Soundness:** 3
**Presentation:** 3
**Contribution:** 3
**Rating:** 8
**Confidence:** 2

**Summary:**

This paper introduces TelecomTS, a large-scale observability dataset from a 5G telecommunications testbed, aiming to fill a critical gap in public time series datasets used for evaluating representation learning in the context of observability. TelecomTS features de-anonymized, heterogeneous covariates with explicit scale information and supports various downstream tasks. Through extensive benchmarks, the authors show that even powerful time-series foundation models and LLMs perform poorly in this regime, rationalizing the unique, under-explored challenges.

**Strengths:**

**Fills a substantial gap in public benchmarks**
TelecomTS addresses the scarcity of openly available, realistic observability datasets by releasing a resource with explicit scale, real and synthetic anomalies, and a range of tasks.

**Comprehensive downstream tasks**
The dataset supports not just classic time-series tasks, but also anomaly detection, anomaly duration localization, root cause analysis, and question-answering tasks.

**Extensive description and explanation of the data curation process**
The paper carefully documents the construction of the dataset, from 5G network setup and controlled jamming for real anomalies to principles for synthetic anomaly generation and diverse Q&A template construction.

**Weaknesses:**

- No direct comparison between normalized data and raw data with scale information.
- Lack of comparison (conceptual or quantitative) with datasets in other domains, especially regarding temporal dependencies.
- The scale of the QA portion is far from yielding significant comparison results.

**Questions:**

1. Could the authors provide a small-scale direct ablation or controlled experiment quantifying the performance improvement within a single model architecture when the absolute scale is preserved or not?
2. Do authors have a plan to expand the number of QA pairs?

---

> ### Author Response · Authors · 2025-11-21
> **Rebuttal Reviewer PdGG**
>
> We sincerely thank **reviewer PdGG** for their encouraging feedback. We appreciate the recognition of the substantial gap that TelecomTS addresses in terms of observability datasets with explicit scale information, as well as the comprehensiveness of our tasks and the depth of our data collection documentation. Below, we address all your concerns and clarify any misconceptions:
>
> - **No direct comparison between normalized and raw data with scale information (W1, Q1):** We appreciate the suggestion and agree that this would help support our claims on the importance of scale information on our downstream tasks. To address this, we perform a direct ablation on Mantis. Specifically, Mantis encodes scale information in its tokenizer through NME scalar encoders [1]. We remove these encoders so that Mantis is only trained on normalized data through regular z-normalization without any explicit scale information. We report the results below:
>
>
>     | **Task \ Model Variant** | Mantis | Mantis w/o NME for scaling |
>     | --- | --- | --- |
>     | Anomaly Detection (Pr, Rec, F1) | (0.640, 0.800, **0.711**) | (0.362, 0.850, 0.510) |
>     | Root-Cause (Acc) | **0.590** | 0.525 |
>     | Anomaly Duration (Pr, Rec, F1) | (0.8734, 0.9144, **0.8934**) | (0.8033, 0.9704, 0.8790) |
>     | Forecasting (MAE, RMSE) | (**0.4578,** **0.6037**) | (0.5703, 0.8436) |
>
>     As expected, we observe a decrease in performance when scalar information is removed. This highlights the importance of TelecomTS, as it provides explicit scale information and opens up avenues for exploring how to best incorporate such scale information to achieve optimal performance. We will include these results in the camera-ready version.
>
> - **Lack of comparison with datasets in other domains (W2):** We point the reviewer to Appendix A, which includes comparisons with datasets from different domains. Our investigation of datasets in other domains showcases several points:
>     1. **Uniqueness of the observability domain:** The Observability domain, to which TelecomTS belongs, has many unique characteristics not covered in other domains. In fact, the data in such domains are characterized by being zero-inflated and exhibiting abrupt behavior and high-variance dynamics [2]. Appendix A contains detailed comparisons to other datasets and points out our other unique characteristics, including fine-grained resolution, heterogeneous variates, and categorical data, which other datasets lack.
>     2. **Lack/Weaknesses of observability datasets:** While observability is critical across many industries, there are very few public datasets in this domain because the data is often proprietary. As a result, existing datasets are typically anonymized and normalized, which limits their usefulness for downstream tasks such as Q&A and prevents any investigation into how scale information can be incorporated into improving performance across these tasks. In contrast, our dataset is de-anonymized and unnormalized, with explicit scale information included.
>     3. **Lack of multimodal datasets with tasks:** TelecomTS is unique in its comprehensive multimodality and tasks. There are a few multimodal time-series datasets in the observability domain, with researchers having to bootstrap their own data.
> - **The scale of the QA portion is far from yielding significant results (W3, Q2):** We believe there is a misunderstanding here. To clarify, we have 69 types of questions for each time-series sample (covering statistics, network, anomaly detection, among others), not a total of 69 questions. Given the number of samples and question types, this results in a total of 2,208,000 QA pairs. To avoid future confusion, we will update Table 1 to explicitly include this number.
>
> [1] “NuTime: Numerically Multi-Scaled Embedding for Large-Scale Time-Series Pretraining”, TMLR 2024.
>
> [2] "This Time is Different: An Observability Perspective on Time Series Foundation Models”, Datadog 2025.

---

### Comment · Area_Chair_7AEL · 2025-11-28
**Official Comment by Area Chair**

Dear Reviewers,

The discussion phase will end soon. Please take a moment to read the authors’ responses carefully and actively engage in the discussion with the authors and your fellow reviewers.

Thanks for your efforts and contributions to ICLR 2026.

Best regards,

Your AC

---

### Meta-Review · Area_Chair_cTzb · 2025-12-10

**Summary:**

This paper introduces TelecomTS, a large-scale real-world observability dataset based on a 5G testbed. TelecomTS comprises heterogeneous, de-anonymized covariates and explicitly includes scale information, supporting a range of downstream tasks such as anomaly detection, root cause analysis, and question-answering benchmarks requiring multimodal reasoning.

Reviewers PdGG and aULz have given highly positive evaluations of this paper, commending its rigorous data collection process, comprehensive task suite, and the significant gap this dataset fills in the field of observability. They found the authors’ rebuttal—including new ablation studies on scale information and the addition of a simple baseline (DLinear)—convincing.

Reviewer uXMG, however, argued for rejecting the paper, citing concerns regarding the dataset’s scale and data quality. The reviewer contended that, given the manually constructed environment and the nature of the dataset, the focus should be on downstream network congestion control and anomaly remediation rather than merely on prediction. After one round of author response, the reviewer maintained their original score.

The final predicted score is 862, indicating a borderline paper. Notably, the positive scores (8 and 6) were given with relatively low confidence levels (3 and 2). Having carefully reviewed the paper, I agree that the authors should further elaborate on the dataset collection process and provide transparent data statistics to clearly demonstrate the paper’s contributions. Additionally, evaluations across a broader range of tasks would be beneficial.

**Reviewer Concerns:**

### Reviewer Concerns Addressed

Practical Value of Scale Information: In response to reviewers PdGG and aULz's inquiry about the empirical benefit of unnormalized data, the authors have provided a dedicated ablation study.

Extended Baseline Models: Addressing reviewer aULz's request for simpler benchmarks, the authors have included results from the DLinear model and KPI-based linear regression models.

Clarification on QA Dataset Size: The initial concern from reviewers PdGG and uXMG regarding the limited number of examples in Table 1 has been clarified. The authors explained that the table illustrates question types, while the actual dataset contains over 2 million QA pairs.

Clarified Role of LLMs: The authors have resolved reviewer aULz's confusion by clearly distinguishing between the use of LLMs for generating support tickets and their separate role in the evaluation process.

### Remaining Reviewer Concerns

Data Quality Inferred from Model Performance: Reviewer uXMG maintains that the modest performance of baseline models suggests potential "poor data quality" or excessive noise. The authors counter that this instead reflects the inherent complexity of real-world observability tasks—characterized by challenges such as zero-inflation and high variance—and validates the benchmark's difficulty. This point remains a point of contention.

Relevance of Forecasting Tasks: Reviewer uXMG questions the relevance of forecasting to the domain stated in the paper, viewing it as inconsistent with the introduction. The authors argue that forecasting is a standard community benchmark essential for demonstrating the unique challenges posed by this specific data distribution.

**Reviewer Scores:**

Reviewer PdGG: Likely to maintain its high score (8). The reviewer has previously given a positive evaluation.

Reviewer aULz: Explicitly indicated a possible increase in the score.

Reviewer uXMG: Maintains its low score. The reviewer stated that its overall evaluation remains unchanged and continues to be skeptical of the dataset's practicality and value.

Therefore, the predicted final score is 862.

---

### Decision · Program_Chairs · 2026-01-26

Reject